



# A climatology of polar stratospheric cloud composition between 2002 and 2012 based on MIPAS/Envisat observations

Reinhold Spang[1], Lars Hoffmann[2], Rolf Müller[1], Jens-Uwe Grooß[1], Ines Tritscher[1], Michael Höpfner[3], Michael Pitts[4], Andrew Orr[5], and Martin Riese[1]

[1]Forschungszentrum Jülich, Institut für Energie und Klimaforschung, Stratosphäre, IEK-7, Jülich, Germany
[2]Forschungszentrum Jülich, Jülich Supercomputing Centre, Jülich, Germany
[3]Karlsruhe Institut für Technologie, Institut für Meteorologie und Klimaforschung, Karlsruhe, Germany
[4]NASA Langley Research Center, Hampton, VA, USA
[5]British Antarctic Survey, Cambridge, UK
*Correspondence to*: R. Spang (r.spang@fz-juelich.de)

**Abstract.** The Michelson Interferometer for Passive Atmospheric Sounding (MIPAS) instrument onboard the ESA Envisat satellite operated from July 2002 until April 2012. The infrared limb emission measurements provide a unique dataset of day
and night observations of polar stratospheric clouds (PSCs) up to both poles. A recent classification method for PSC types in IR limb spectra using spectral measurements in different atmospheric window regions has been applied to the complete mission period of MIPAS. The method uses a simple probabilistic classifier based on Bayes' theorem with a strong independence assumption on a combination of a well-established two-colour ratio method and multiple 2D probability density functions of brightness temperature differences. The Bayesian classifier distinguishes between solid particles of ice, nitric acid trihydrate
(NAT), liquid droplets of super-cooled ternary solution (STS) as well as mixed types.

A climatology of MIPAS PSC occurrence and specific PSC classes has been compiled. Comparisons with results from the classification scheme of the spaceborne lidar Cloud-Aerosol Lidar with Orthogonal Polarization (CALIOP) on the Cloud-Aerosol-Lidar and Infrared Pathfinder Satellite Observations (CALIPSO) satellite show excellent correspondence in the spatial and temporal evolution for the area of PSC coverage ($A_{PSC}$) even for each PSC class. Probability density functions of the PSC
temperature, retrieved for each class with respect to equilibrium temperature of ice and based on coincident temperatures from meteorological reanalyses, are in accordance with the microphysical knowledge of the formation processes with respect to temperature for all three PSC types.

This paper represents unprecedented pole-covering day and night time climatology of the PSC distributions and their composition of different particle types. The dataset allows analyses on the temporal and spatial development of the PSC
formation process over multiple winters. At first view, a more general comparison of $A_{PSC}$ and $A_{ICE}$ retrieved from the observations and from the existence temperature for NAT and ice particles based on ECMWF reanalysis temperature data



show the high potential of the climatology for the validation and improvement of PSC schemes in chemical transport and chemistry climate models.

## 1    Introduction

The essential role of polar stratospheric clouds (PSCs) in the depletion of stratospheric ozone has been well established (e. g. Solomon, 1999). Heterogeneous reactions on PSC particles convert chlorine reservoir species (in particular HCl and ClONO$_2$) to chlorine radicals that destroy ozone catalytically. Although PSCs have been explored now for more than 30 years, there are still many open questions that limit our ability to accurately simulate the formation and surface area of different PSC types and consequently, limit the quality of the prediction of future polar ozone loss rates in a changing climate system (e.g. Peter

and Grooß, 2012). For example, although the closure of the Antarctic ozone hole is projected to occur by the end of twenty first century, large uncertainties exist in the rate and timing of its recovery (e.g. Eyring et al., 2013, WMO, 2014, Fernandez et al., 2017).

PSCs are located in the cold polar vortices in both winter hemisphere, and are classified into three types: Super-cooled ternary solution droplets (STS), nitric acid hydrates (most likely nitric acid trihydrate, NAT), and ice particles (e.g. Lowe and

MacKenzie, 2008).  Their formation and existence is mainly determined by the temperature with an ice frost point ($T_{ICE}$) at ~188 K and an equilibrium NAT temperature ($T_{NAT}$) of ~196 K (at 55 hPa, for 5 ppmv H$_2$O, and 10 ppbv HNO$_3$). Due to nucleation barriers temperatures lower than the existence temperatures are required. A possible nucleation pathway for NAT needs pre-existing ice particles and therefore temperatures 3-4 K below $T_{ICE}$. STS droplets grow with decreasing temperatures from binary to ternary solution droplets. Temperatures, where the HNO$_3$ weight percentages are as large as H$_2$SO$_4$ and the

volume increases strongly, are described with $T_{STS}$ (~192 K). The formation of NAT requires much lower temperatures, usually 3–4 K below the ice frost point on pre-existing ice particles or at higher temperatures on meteoritic dust particles (e.g. Peter and Grooß, 2012, Hoyle et al., 2013). The formation of STS droplets is well understood (e.g. Carslaw et al., 1995), but for NAT and ice particles, new formation mechanisms by heterogeneous nucleation on meteoric smoke at higher temperatures are under discussion (e.g. Hoyle et al., 2013; Engel et al., 2013; Grooß et al., 2014).

PSCs impact polar chemistry in three ways: (1) through heterogeneous reactions, where the rates depend on particle surface area and particle concentration (Sander et al., 2011, Shi et al., 2001); (2) through the uptake of HNO$_3$ into PSCs thereby reducing HNO$_3$ concentrations in the gas-phase temporarily (e.g. Salawitch et al., 1989); (3) NAT particles can grow to large enough sizes for sedimentation, so that a permanent removal of the gas phase HNO$_3$ from polar air masses ('denitrification') can occur (e.g. Fahey et al, 2001, Molleker et al., 2014) and has a significant impact on polar ozone loss in the Arctic (Waibel

et al., 1999).





Difficulties in making accurate predictions on the ozone recovery in a changing climate stems from a variety of problems in chemistry-climate models (CCMs). One important problem is the poor representation of PSCs. CCMs used for assessments of polar stratospheric ozone loss (e.g., Eyring et al., 2013) often employ rather simple PSC schemes. Although, these simple schemes today mostly include reactions on all types of PSCs, the simplifications may lead to a NAT-dominated heterogeneous

chemistry. However, it is known that heterogeneous chemistry on super-cooled ternary solution and on cold binary aerosol particles probably dominates polar chlorine activations (e.g., Solomon, 1999; Drdla and Müller, 2012, Kirner et al., 2015). The models usually do not include comprehensive microphysical modules to describe the evolution of different types of PSCs over the winter. Additional, mesoscale temperature variations caused by orographic gravity waves can be crucial for the formation of PSCs, especially for conditions close to the temperature threshold for particle formation (Carslaw et al., 1998, Dörnbrack

et al., 2002, Engel et al., 2013, Hoffman et al., 2017), but are missing from the current generation of CCMs (Orr et al., 2015). Assumptions on the occurrence of different types of PSCs in CCM simulations have been shown to only have a limited impact on many aspects of polar ozone loss in case studies for both the Arctic and Antarctic. For example, liquid PSC particles alone are sufficient to simulate nearly all of the ozone loss using current model chemistry (Wohltmann et al., 2013; Kirner et al., 2015; Solomon et al., 2015). However, which type of PSs is present at the top of the ozone loss region (between 10 and 30

hPa) is important, as shown for Antarctica by Kirner et al. (2015). Further, in the Arctic polar vortex of the 2009/2010 winter the initial activation occurred in PSCs covering only a small portion of the vortex; under such conditions the type of PSC present can be decisive (Wegner et al., 2016).

Heterogeneous reaction rates on PSCs and cold binary aerosols are always strongly temperature dependent, but the reaction rate for a particular heterogeneous reaction also depends on the PSC particle type (e.g., Drdla and Müller, 2012, Wegner et al.,

2012). Further, although commonly faster rates are assumed, there is still a substantial uncertainty on the rates of heterogeneous reactions on NAT (Carslaw et al., 1997, Wegner et al., 2012) which makes determining the type of PSC present in the atmosphere important.

Consequently, measurements of the particle type of PSCs are highly desirable but are so far only very limited. Ideally, these measurements should cover the complete polar vortex and should last for multiple winters, which is only possible either

passively in the mid-infrared or actively through lidars measurements from space. Climatologies of PSC observations and their classifications in PSC types over several winters are so far restricted to ground based stations - usually lidar systems (e.g. Di Liberto et al., 2014, Achtert and Tesche, 2014), or are based on spaceborne solar occultation measurements (e.g. Hervig et al. 1997, Fromm et al., 2003) usually without or only very limited PSC type information (Strawa et al., 2002). Consequently, these datasets are restricted to local sites or are directly attached to the solar terminator, and therefore do not allow

measurements inside the cold and dark winter polar vortices. UV/vis limb scattering measurements can also measure PSC distributions (von Savigny et al., 2005), but currently do not allow PSC types to be discriminated and are restricted to daylight




conditions. Stellar occultation measurements in the UV/vis wavelength region can partly compensate for the difficulty of adequate coverage of the polar night cap (e.g. Vanhellemont et al., 2010) but are so far not capable to differentiate PSC types. Limb infrared emission measurements have the advantage of being available at day and night time which allows unrestricted sounding of both polar vortices (e.g. Spang et al. 2001). With a certain spectral resolution of the measurement it is even possible to discriminate between specific PSC types (Spang and Remedios, 2003, Spang et al., 2004). The restricted time series of polar winter measurements by the Improved Stratospheric and Mesospheric Sounder (ISAMS) (Taylor et al., 1994), the Cryogenic Limb Array Etalon Spectrometer (CLAES) (Massie et al., 1994), and the Cryogenic Spectrometers and Telescopes for the Atmosphere (CRISTA) (Spang et al., 2001) already showed the high potential of infrared (IR) limb measurements in the field of PSC research. A more complete picture of PSC morphology and composition on polar vortex-wide scales is emerging from a suite of recent satellite missions. The Michelson Interferometer for Passive Atmospheric Sounding (MIPAS) (Fischer et al., 2008) on Envisat (2002-2012) and the Cloud-Aerosol Lidar with Orthogonal Polarization (CALIOP) (Winker et al., 2009) on CALIPSO (2006-present) retrieve PSC information beyond comparison on horizontal, vertical, and temporal coverage and resolution from space. These datasets have already motivated numerous PSC studies that both extend and challenge our present knowledge of PSC processes and modelling capabilities (e.g. Peter and Grooß, 2012, and references therein). Consequently, one of the main objectives of the actual Stratosphere-troposphere Processes And their Role in Climate (SPARC) PSC activity is to synthesize and archive these new datasets in combination with former in-situ balloon borne (Antarctic and Arctic) and remote lidar (Antarctic) measurements into a state of the art PSC climatology. Here, we present one part of the planned combined PSC climatology, comprising a new MIPAS PSC dataset covering the full mission lifetime of ten years from July 2002 to March 2012.

The paper is structured as follows. Section 2 introduces MIPAS and other involved instruments. Section 3 provides a summary of the MIPAS detection and classification method - more extensively presented in Spang et al. (2016) - is given in conjunction with a comparison to CALIPSO and plausibility tests for the results of the classification approach. Section 4 describes in detail various aspects of the climatology, like temporal and spatial evolution of the PSC type distribution. In addition, first analyses of retrieved parameters from the climatology which are important for the evaluation of CCM/CTMs are presented, for example time series of the area of the hemisphere covered by PSCs ($A_{PSC}$) and the three corresponding PSC types ($A_{ICE}$, $A_{NAT}$, $A_{STS}$).

## 2 Instruments and datasets

### 2.1 MIPAS instrument on Envisat

The MIPAS instrument on board the Envisat satellite measured limb infrared spectra in the wavelength range from 4 to 15 μm (Fischer et al., 2008) from July 2002 to April 2012. The satellite operated in a sun synchronous orbit (inclination 98.4°) and allowed geographical coverage up to both poles due to additional poleward tilt of the primary mirror (usually up to 87°S and 89°N). Due to technical problems with the interferometer in 2004 the exceptionally high spectral resolution of 0.025 cm$^{-1}$



(HR: high-resolution mode) was reduced to 0.0625 cm$^{-1}$ (OR: optimised-resolution mode) (Raspollini et al, 2013). The processed level 1b radiance data from the measurement period July 2002 to March 2004 (phase 1) were measured in the HR mode and from January 2005 to March 2012 (phase 2) in the OR mode. Also the vertical and horizontal sampling for the nominal measurement modes were changed from phase 1 to 2. A fixed vertical step size of 3 km up to a tangent height of ~42 km was used in phase 1 and changed to an altitude dependent vertical step size of 1.5 – 4.5 km for phase 2, with increasing

altitude steps with height. In addition, the lowest tangent altitude was changed from a constant value to latitude-dependent values of 5 km at the poles up to 12 km over the equator in phase 2. The horizontal sampling was improved from 550 km to 420 km.

The trapezoidal form of the vertical field of view (FOV) of MIPAS has a base width of 4 km and a top width of ~2.8 km. In cross track direction the FOV covers a range of 30 km. In the following analyses Version 5 level 1b data, processed and

provided by European Space Agency (ESA), have been used as the underlying dataset for cloud detection and classification of the PSCs by composition. More details on detection and classification are given in Section 3.

### 2.2 CALIOP instrument on CALIPSO

The Cloud-Aerosol Lidar with Orthogonal Polarization (CALIOP) instrument is a dual wavelength polarization-sensitive lidar

that provides high vertical resolution profiles of backscatter coefficients at 532 and 1064 nm (Winker et al., 2009). CALIOP is the primary instrument on the CALIPSO (Cloud-Aerosol-Lidar and Infrared Pathfinder Satellite Observations) satellite, which flies in a 98° inclination sun-synchronous orbit at an altitude of 705 km. This orbit geometry facilitates nadir-viewing measurements up to latitudes of 82°N/S. CALIOP PSC analyses are based on nighttime-only CALIPSO Level 1B 532-nm parallel and perpendicular backscatter coefficient measurements smoothed to a uniform 5-km horizontal (along track) by 180-

m vertical grid over the altitude range from 8.2-30 km.

This study uses data from the CALIPSO Lidar Level 2 Polar Stratospheric Cloud Mask Version 2.0 (Poole and Pitts, 2017), in which PSCs are detected as statistical outliers relative to the background aerosol population in either 532-nm perpendicular backscatter coefficient or 532-nm scattering ratio (the ratio of total to molecular backscatter). To facilitate detection of optical thin clouds, successive horizontal averaging (5, 15, 45, 135 km) is applied to the data to improve signal-to-noise ratio (SNR)

(Pitts et al., 2009). Based on comparison of measurements with theoretical optical calculations for non-equilibrium liquid-NAT or liquid-ice mixtures, CALIOP PSCs are classified by composition into five categories: STS, (liquid-) NAT mixtures, enhanced NAT mixtures (those liquid-NAT mixtures with very high NAT number densities), ice, and wave ice (ice PSCs induced by mountain waves). Version 2.0 CALIOP PSC data is significantly improved over the original version, especially in the separation of NAT mixtures and ice in the presence of denitrification and dehydration.




## 3 PSC detection and classification

The PSC detection and classification approach used for MIPAS is described in detail in Spang et al. (2012), which presents the MIPclouds processor for detection and cloud parameter retrievals, and in Spang et al. (2016), which also introduces the methodology of the Bayesian classifier for PSC types and represents the results of the classification in comparison to the CALIOP classification. In the following, we only summarise the major principles and characteristics of the applied methodology.

### 3.1 MIPAS detection of PSCs

The MIPclouds processor and its data products are used for the detection of cloudy spectra in each profile of the measurement period. A step-like data processing approach of up to 5 detection methods was chosen for the processor to provide summary cloud top height (CTH) information with the best possible detection sensitivity (Spang et al., 2012). The most common detection method for IR limb measurements is the cloud index (CI) colour ratio approach with constant detection threshold (Spang et al., 2001, Spang et al., 2004), but also variable CI threshold profiles changing with altitude, and with altitude, latitude, and month (Spang et al., 2012, Sembhi et al., 2012) were combined.

The CI is a ratio of the mean radiances at a wavelength of 796 $cm^{-1}$, dominated by $CO_2$ emissions, and around 832 $cm^{-1}$, an atmospheric window region. The CI has been extensively used for the detection of cloud contaminated spectra for various types of IR limb measurements of space and airborne instruments (e.g. Spang et al., 2001, Spang et al., 2004, Spang et al., 2007) and has been established as a simple, robust and, depending on the applied threshold value, highly sensitive cloud detection method. The CI is well suited for the detection of PSCs, because it correlates extremely well with the integrated particle volume or area density along the limb path (VDP/ADP), and also with optical depth, extinction. ADP or VDP are the integrated particle size distribution quantities where the IR limb signal of a cloud is most sensitive to (Spang et al., 2012). The long path length through the atmosphere enables to detect extremely low volume and area densities if the cloud has a certain horizontal extent (several kilometres). For example, a 1 km or 100 km horizontally extended ice cloud is detectable for ice water contents of $> 0.3$ mg/m$^3$ and $> 0.003$ mg/m$^3$, respectively (Spang et al., 2015).

In addition, a retrieval approach with simplified assumptions for the radiative transfer in clouds (Hurley et al., 2011) was implemented in the MIPclouds detection algorithm. This retrieval copes with the difficulty of using a 3 km vertical FOV to determine a more realistic cloud top height (CTH) inside the FOV. We use the step-like approach of all detection methods and the summary CTH for selecting the first cloud affected spectrum in an altitude scan for starting with the PSC classification for this and all potentially cloudy tangent heights below.



### 3.2 Bayesian classifier for IR limb measurements

**Figure 1** shows a schematic overview of the Bayesian classifier approach applied in this study. Basis for the classification is an extensive database of more than 600,000 modelled PSC spectra (Spang et al., 2012) with varying PSC types: ice, NAT and STS ($n_{type}$=3), with variable micro-physical (volume density, median radius of the particle size distribution) and macro-physical (horizontal and vertical extent) parameters for the modelled cloud scenes. Mean radiances of seven wavelength regions, mainly atmospheric window regions, where cloud emissions can contribute significantly to the measured signals, were selected for

the computation of two colour ratios, cloud index (CI) and so-called NAT index (NI), as well as five brightness temperature differences (BTD).

The NI is a colour ratio of the 820 cm$^{-1}$ to the 796 cm$^{-1}$ wavelength region, with the latter region already used in the CI approach. It is a measure for enhanced peak-like emission at ~820 cm$^{-1}$ by a characteristic spectral feature in the NAT emission spectrum (Höpfner et al., 2002, Spang and Remedios, 2003). These emissions are strongly enhanced and are clearly detectable

in NAT cloud spectra for median radii r < 3 µm (Höpfner et al., 2006a). The combination of CI and NI, like illustrated in the probability density function (PDF) (I) in **Figure 1,** was the first sophisticated method to attribute IR limb spectra to specific PSC types. The three highlighted regions (1) to (3) separate ice in region (1) from NAT with r < 3 µm in region (2) and most likely STS, but also large NAT particle or optically very thin ice clouds in region (3). The CI-NI approach was first applied by Spang et al. (2003) to the CRISTA southern polar winter observations in August 1997. Although the method has some

weaknesses in the differentiation of ice and STS as well as difficulties to discriminate between cloud emissions in the IR spectra dominated by STS particles with emissions by large NAT particles (r > 3 µm), the method has been used and improved in various studies and with different instruments (Spang et al., 2005a/b, Höpfner et al., 2006a/b, Arnone et al., 2012, Lambert et al., 2012).

Measurements of large NAT-particles, so-called NAT-rocks, are of specific interest because it is the particle type causing

denitrification in the polar vortex (Fahey et al., 2001, Molleker et al., 2014). Measurements - in situ as well as remote sensing - of these particle are very difficult and quite rare in the literature. In addition, the formation process of NAT and NAT-rocks in the polar stratosphere is still under debate (e.g. Peter and Grooß, 2012) and consequently, more sophisticated information on PSC particle type and size are desirable.

To improve the classification and to bypass specific weaknesses of the CI-NI approach in total four classification diagrams

($i$=1...$m$, with $m$=4) have been selected by Spang et al. (2016). These are combining different sensitivities to cloud composition and optical thickness (depth) of the clouds and are retrieved from the cloud scenario database of the MIPclouds study (Spang et al., 2012) and the MIPAS measurements themselves (e.g., the applicable range in the parameter space of the correlation diagrams is guided by the PDF of the measurements). All classification diagrams show distinctive regions (in total 13) which can be attributed to a specific composition of PSCs or potentially mixtures of PSC types. The four methods are then combined

in a simple probabilistic classifier based on applying Bayes' theorem with strong (naive) independence assumptions. Based on





the radiative transfer calculations covering the full parameter space of PSCs in the database and the MIPAS measurements themselves, a-priori-like probabilities are defined for each classification region. Examples of monthly probability density functions (PDFs) of the MIPAS measurements for each classification diagram are presented in **Figure 1** for July 2010.  For a single spectrum the normalised product probability ($P_j$, with $j = 1 ... n$ and $n=3$) for each PSC type is computed and step-wise

decision criteria are defined:

  (1) **if** the maximum of $P_j$ (with $j=1...3$) $> 50\%$ **then** Type ($j$) is the most likely PSC composition;

  (2) **if** two $P_j$'s are between 40 and 50% **then** a mix-type class, for example ICE_STS for j=1 and j=3, is attributed;

  (3) **if** the classification results in $P_1$, $P_2$, $P_3 < 40\%$ **then** the cloud spectrum is unclassified and pigeon-holed to the class *unkown.*

Analyses with the Version 1.2.8 Bayesian Classifier for MIPAS PSC observation showed that the mix-type classes ICE_STS and ICE_NAT have an insignificant partitioning (<0.1% per month) with respect to all cloud classes, and are negligible for further statistical analyses (Spang et al., 2016). Whereby, the STS_NAT class can show larger occurrence rates (5-10% per month) especially for the SH early winter conditions (**s. Sec. 3.5**).

Spang et al. (2016) demonstrated the overall reliability of the results of the new classification method by detailed sensitivity

tests on the Bayesian classifier input parameter and extensive comparisons with the CALIOP lidar classification. The latter comparison worked on a large statistical basis with narrow coincidence criteria and showed an overall good agreement between the partitioning of the defined PSC classes but also some significant miss-matches. Miss-matches are highlighting differences between the two instruments MIPAS and CALIOP with respect to the detection sensitivity for specific PSC types (and mixtures of them) due the very different measurement technique and geometry (Spang et al., 2016).

It should be noted that MIPAS and CALIOP have completely different sensitivities to measure cloud structures with a mixture of PSC types. It is quite plausible that PSCs will usually include mixtures of all three particle types (ice, NAT, and STS) (Pitts et al., 2013) dependent on the temperature and available trace gas concentrations ($HNO_3$ and $H_2O$), but the intensity of the analysed measurement signals, for example NI for MIPAS or the backscatter ratio for CALIOP, can have quite different sensitivities to detect one specific type in a mixture of PSC types in the measurement volume. Limb measurements are

characterised by a horizontally extended measurement volume (hundreds of kilometres). The line of sight will usually pass various regions dominated by different particle types. Consequently, the effect of mixtures are very difficult to quantify and are only partly considered in the PSC classification of MIPAS and CALIOP.

**3.3 Retrieved parameter and data processing**

The main macro-physical cloud retrieval parameter is the cloud top height information and a cloud mask of flags, where for each cloudy spectrum at and below the CTH and down to tangent heights of 12 km the classification result is presented. For the Version 1.2.8 Bayesian classifier the following cloud mask of PSC classes is defined: -1: unclassified (none-cloudy), 0:





unknown, 1: Ice, 2: NAT, 3: STS, 4: ICE_NAT, 5: STS_NAT, and 6: ICE_STS. The classification is only applied to spectra with CI < 6 at or below the first tangent height level of detection. This more stringent CI threshold compared to the CI-based

detection methods used step-like detection algorithm of MIPclouds gives the opportunity to classify spectra which are most likely cloudy but are only detected by the most sensitive methods and may be missed in the combined MIPclouds detection approach - mainly optically very thin cloud layers (Spang et al., 2012). This may introduce some biases in the statistics by false-positive events. However, this risk occurs only at the lower bound of the cloud layer (cloud base height) and for detections close to the signal to noise level (altitudes > 30 km). These can be easily be excluded by a restriction in altitude from the

following analyses.

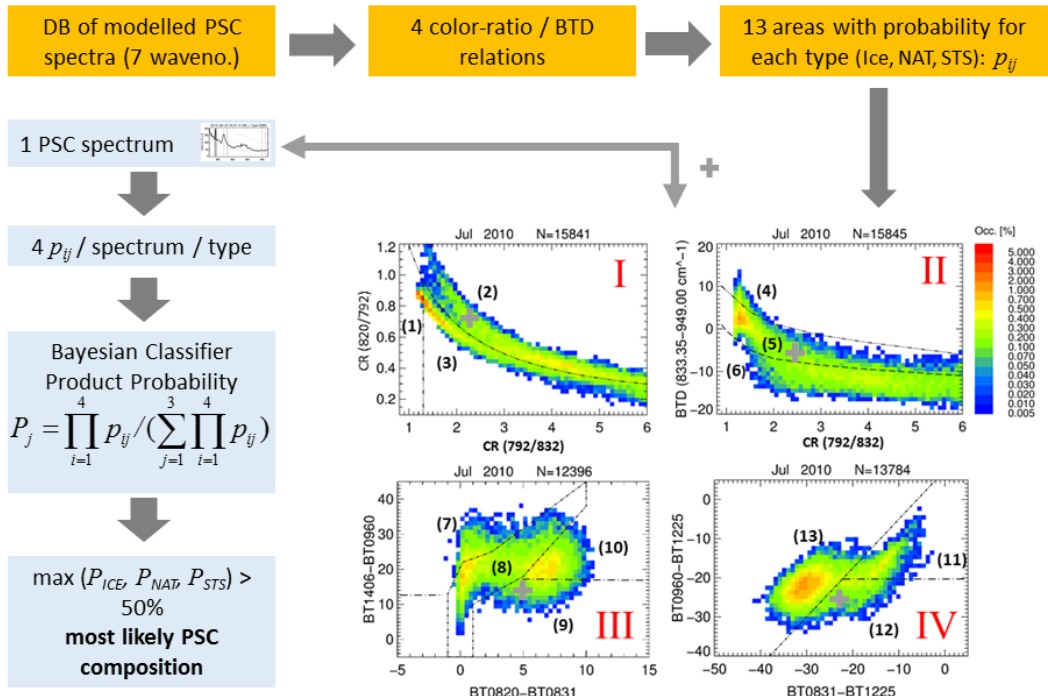

**Figure 1:** Schematic diagram of the Bayesian classifier procedure (blue boxes) for the MIPAS PSC type classification. Orange boxes illustrate the part of the huge database (DB) of modelled PSC spectra for the computation of colour rations and brightness
temperature differences (BTD). The diagrams I to IV are MIPAS examples of monthly probability density functions (PDF) for SH July 2010 data applied in the Bayesian classifier. The grey cross in each of the four PDF diagrams is illustrating the input information of a single PSC spectrum, which is used to retrieve the most likely PSC composition (for details see text and Spang et al., 2016).





There is a general caveat to separate emissions in the IR by large NAT particles from emissions by STS particles due to their similar behaviour (Spang et al., 2016). Consequently we cannot exclude larger abundance of large NAT particles (r > 3 μm) in cloud events classified with STS by the Bayesian classifier, and the term STS$_{mix}$ is probably a better description for this type of MIPAS PSC class. Unfortunately, there is no proper approach to quantify or validate this uncertainty. For example, the CALIOP classification has also weaknesses in differentiation of large NAT particles from small NAT and STS particles

adequately, which does not allow us to use CALIOP as a transfer standard for MIPAS.

Additional meteorological information on the coincident temperature, pressure and potential temperature based on ERA-interim data (ERAi) (Dee et al., 2011) are also merged into the dataset for simplicity of statistical analyses with the PSC dataset. Additional information on the geographical location, time, and satellite position per spectrum are part of the data structure as well as parameter retrieved from MIPAS measurements and used in the detection and classification algorithm (e.g.

CI and NI for each measured spectrum) and are saved in the Network Common Data Format (NetCDF) output files of the data processing scheme. Details on the set of parameter stored in the output files (PSC related and auxiliary parameter) are presented in **Sec. 6**.

The complete Envisat mission with MIPAS data from July 2002 to April 2012 has been processed with the Bayesian classifier. There are data gaps between July and September 2007, and extended data gaps up to months of no data between April and

December 2004 followed by a period with reduced temporal coverage between January 2005 and December 2005. During this period, the repeat cycle of measurements per week was increased in a stepwise process from 50% back to 100%. Afterwards a very homogeneous spatial and temporal coverage has been achieved on the loss of contact to the transmission of Envisat on 8 April 2012. The resulting ten winter seasons in the northern hemisphere (2002/2003 to 2011/2012) and nine seasons in the southern hemisphere (2002, 2003, and 2005 to 2011) represent a unique dataset of day and night time PSC type measurements

up to the poles for process studies in the polar winter stratosphere. This is especially important for the evaluation and improvements of CCMs and CTMs for a better predictability of long term ozone trends in a changing climate.

### 3.4  Examples of the Bayesian classifier results

The usefulness of the new PSC classification algorithm has already been presented in Spang et al. (2016) and Hoffmann et al.

(2017). These studies showed that some of the MIPAS PSC detections are directly linked to small-scale temperature fluctuations due to gravity waves (GW). **Figure 2** shows two examples for the horizontal distribution of the PSC classes of the Bayesian classifier for a single day in January 2010 for the northern hemisphere (NH) and June 2011 in the southern hemisphere (SH) in an altitude region defined by the potential temperature range $\Theta$ = 500 K ± 20 K. Both examples highlight very special events and characteristics of the classifier dataset. The NH example shows an exceptionally large synoptic area

of ice clouds north of Scandinavia surrounded by the usually more frequent observations of NAT and STS clouds in this region, as temperatures are usually significantly higher than T$_{ICE}$. Note that the Montgomery stream function contour lines suggest




that downstream of this region NAT particles exist, which have been 'seeded' in the ice cloud area and may formed on pre-existing ice.

The example for the SH in June 2010 shows a typical PSC type distribution for this time of the winter, which involves the
wide spread occurrence of NAT particles usually around mid of June. The map shows indications for the development of the so-called NAT-belt (Tazadabeh et al., 2001). Höpfner et al. (2006b) and Eckermann et al. (2009) showed that this kind of distribution is mainly composed of small NAT particles (r < 3 µm, which are triggered by strong mountain wave (MW) activity over the Antarctic Peninsula. The Antarctic Peninsula is a well-known hot spot region for mountain waves (e.g. Wu and Jiang, 2002, Hoffmann et al., 2017), and the link between specific mountain waves events and the NAT formation process have been
analysed by Hoffmann et al. (2017) in more detail with MIPAS dataset and Atmospheric Infrared Sounder (AIRS) GW observations. The study of Hoffmann et al. found many events (>50), where the PSC occurrence was clearly linked to MW activity. There are additional MW source regions in Antarctica where this link is also observed, e.g. the Transantarctic Mountains, but the Antarctic Peninsula is by far the most prominent region. MWs may play a prominent role in the total abundance of PSCs, especially in the NH where the temperatures are more often close to but not significantly below the
threshold temperatures for PSC formation. Consequently, the link between MWs and GW sources in general needs special emphasis in CCMs and CTMs for a better accuracy in the prediction of the future polar ozone in a changing climate (Orr et al., 2014).

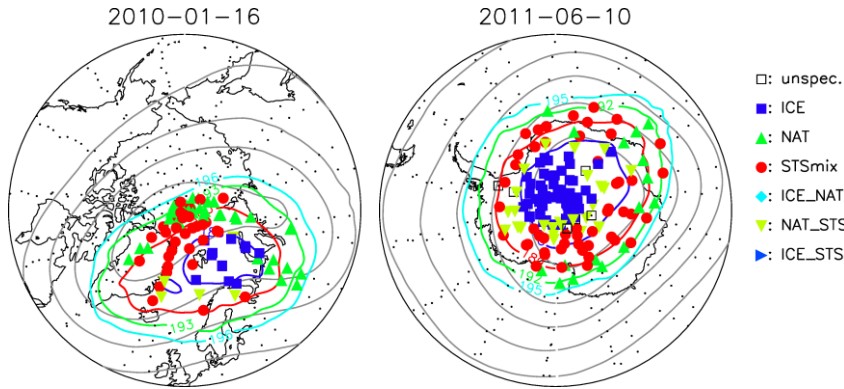


**Figure 2:** Examples of the horizontal distributions of MIPAS PSC composition classes for NH winter and SH winter conditions at an altitude level of 500 K potential temperature (± 20 K). In addition to the different cloud type symbols described in the figure legend (see main text for details), temperature contours for $T_{ICE}$ (dark blue) (Marti and Mauersberger, 1993), $T_{STS}$ = ($T_{NAT}$+ $T_{ICE}$) / 2 (red), $T_{NAT}$ (green) (Hanson and Mauersberger, 1988), and $T_{NAT}$ + 2 K (turquoise) based on ERA-Interim
data (Dee et al., 2011) with constant water vapour and nitric acid mixing ratios, grey contours of the Montgomery stream function at 500 K, and black dots for non-cloudy MIPAS profile locations are superimposed.



### 3.5 Temperature probability distribution of PSC types

Temperature PDFs of various classified PSC classes are a useful tool for testing qualitatively the reliability of classification
methods (e.g. Pitts et al., 2013) and are extremely valuable for the analysis of PSC formation processes and their chemical
imprint by heterogeneous reaction on the polar ozone layer (Pitts et al., 2013, Lambert et al., 2012). Mean SH and NH winter
histograms are presented in **Figure 3** and **Figure 4** related to $\Delta T = T - T_{ice}$ for all measured PSC spectra between 16 and 30 km
and selected six SH and four NH polar winters, respectively. As well as showing considerable inter-annual variability for both
regions, the PDFs for the southern polar winter differ quite considerably from those for the northern polar winter, for example:

(a) The ice type histograms for the SH show the coldest and narrowest PDFs, with its maxima slightly below or just
at $T_{ICE}$. Arctic winter conditions are usually not cold enough to produce comparable maxima to STS or NAT like
in the Antarctic, but a few winters (e.g. 2009/2010 and 2010/2011) show a certain activity in ice formation.

     (b) The STS class is usually dominating the NH distribution, due to generally warmer vortex conditions. The
temperature of the maximum of the PDF distribution for STS is ~3-4 K above $T_{ICE}$, which is in line with the
equilibrium temperatures of STS and ice (e.g. Carslaw et al., 1995). However, for specific meteorological
conditions like in winter 2009/2010 and 2010/2011 the NAT population has a similar proportion like in the 'warm'
SH winters (e.g. 2010, see also **Sec. 4.1.2**).

     (c) STS and NAT show both very similar PDF shapes for the SH and NH (maximum location, $\Delta T_{max}$ and width) but
a slightly shifted PDF maximum for NAT to $T - T_{ICE} > 5$ K in the NH. The latter maximum might be affected by
the rare occasions of temperatures below $T_{ICE}$ in the NH. Therefore heterogeneous NAT nucleation on pre-
existing ice particles can only dominate the NAT formation in the SH, whereby in the NH heterogeneous NAT
formation at $T > T_{ICE}$ may play a dominant role. Recent analyses of in-situ and satellite measurements together
with new formation mechanisms in microphysical models support this so far excluded formation pathway at
$T > T_{ICE}$ (Hoyle et al., 2013). Despite the ongoing debate over homogeneous and heterogeneous nucleation, an
analogues analysis by Pitts et al. (2013) for CALIOP data and independent classification scheme shows nearly
identical relations between the STS and the mixed classes of CALIOP (mixtures of STS with contributions of
NAT particles depending on the number densities and particle sizes of NAT).

     (d) The mixed-type class NAT_STS shows very asymmetric PDFs with a shift to the cold-tail of STS (SH/NH) and
NAT (SH). These 'cold tail'-class might indicate the coexistence of STS and NAT particles formed by
heterogeneous nucleation just around $T_{ICE}$. However, IR measurements of small NAT particles usually show the
characteristic NAT spectral feature at 820 cm$^{-1}$ (Höpfner et al., 2002, Spang and Remedios, 2003). PDFs for these
events, selected with the former CI-NI classification approach (s. Sec. 3.2, Spang and Remedios, 2003, Höpfner
et al., 2006a), are highlighted in **Figure 3 and 4** by the black line. PDF$_{CI-NI}$ show no obviously enhanced number
for NAT events at corresponding $\Delta T_{max}$ of the PDF$_{NAT\_STS}$, but a significant enhancement in the centre region of





the PDF$_{NAT}$ (most likely an overestimation). Nevertheless, we cannot exclude from this comparison that PDF$_{NAT\_STS}$ might be part of the PDF$_{CI-NI}$. In addition, the PDF$_{CI-NI}$ shows a significantly larger number of events than the Bayesian classifier-based PDF$_{NAT}$ (~20%). Frequently, very weak signatures of the NAT feature may classified as ice or STS due to characteristic radiance contribution of these types in other wavelength regions. These spectra can appear in regions of the two-parameter scatter distributions II to IV of **Figure 1** with higher

probabilities for STS or ice than for NAT.

Overall, the behaviour in the T-T$_{ICE}$ histograms are in line with the current understanding of the formation of PSCs in the winter polar vortices, which gives us confidence that the Bayesian classifier is a reliable approach providing data for more detailed scientific studies on the formation processes of PSCs as well for a climatology of PSC composition of the entire MIPAS measurement period.

A similar temperature analysis of the PSC classes by Pitts et al. (2013, Fig. 8 and Fig. 9) with the CALIOP lidar data shows comparable results with respect to the PDF maxima location and shift between the PSC types. This is an independent endorsement in the reliability of the new MIPAS approach. u, the CALIOP analysis shows significantly smaller PDF distribution widths than the MIPAS analysis. The typical difficulty for limb sounders to define the exact position of the cloud along the line of sight (~400 km) will raise additional uncertainties in the cloud temperature assignment, where the coincidence

of the tangent point with the ERAi temperature information is applied. Coincident H$_2$O and HNO$_3$ measurements like from the Microwave Limb Sounder (MLS) on the EOS-Aura satellite for CALIOP with virtual zero miss time and miss distance (e.g. Lambert et al., 2012) are used in the CALIOP analyses, but are not available for MIPAS. To compensate for this difficulty, we applied MLS-based zonal mean water vapour measurements in equivalent latitude coordinates to the MIPAS tangent point equivalent latitude. Yet, both approximations for the MIPAS analysis will produce significant noise-like scatter in the

histograms and consequently a broader distribution width than in the CALIOP analyses.





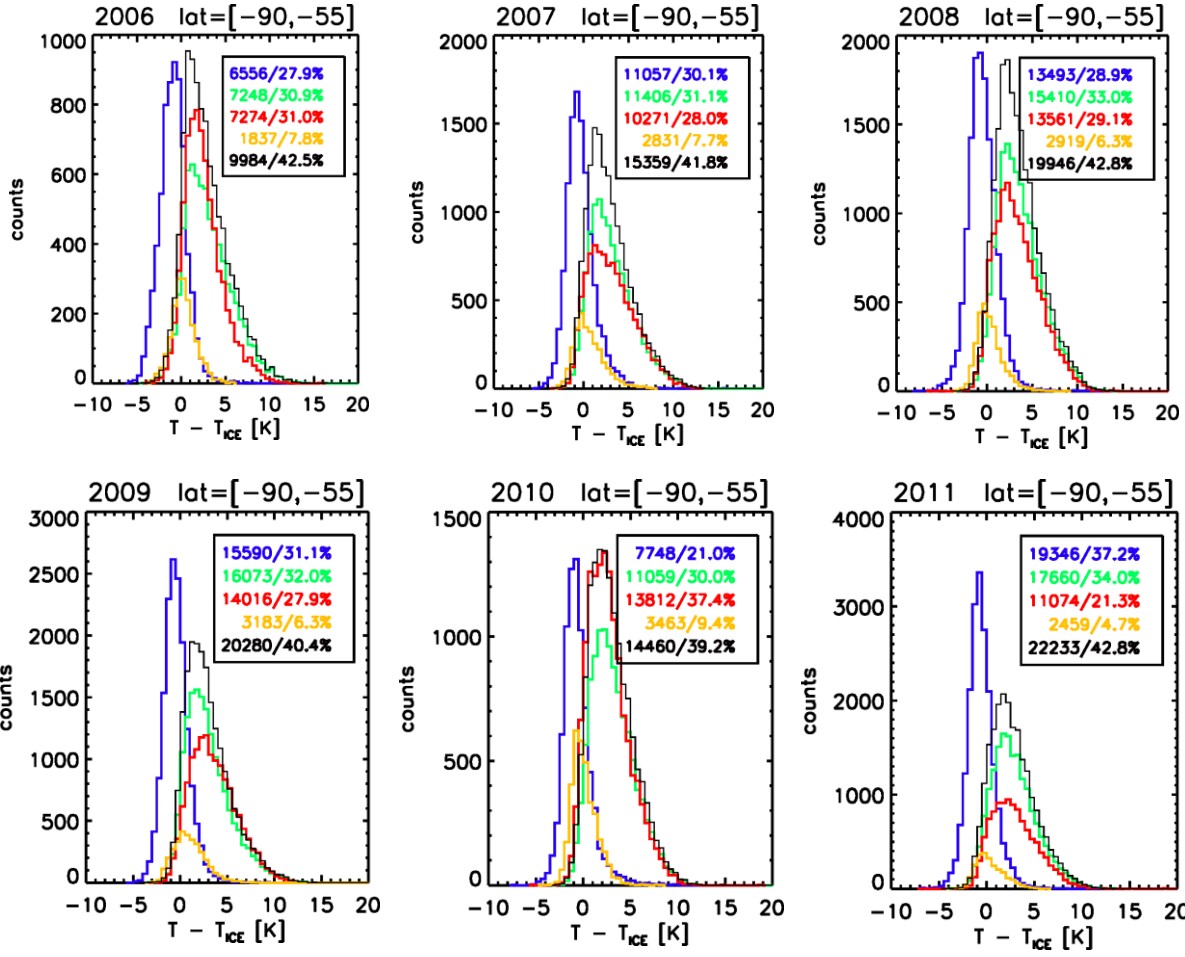

**Figure 3:** Six years of southern polar winter PSC class PDF distributions as a function of T-T$_{ICE}$. For consideration of the dehydration-process in the polar vortex over the winter coincident temperatures from ERA-Interim reanalyses and zonal mean water vapour mixing ratios on equivalent latitudes of the MLS instrument (e.g. Lambert et al, 2012) are used for the computation of T$_{ICE}$ (Marti and Mauersberger, 1993). The years 2006-2011 are presented for the four main PSC classes ice (blue), NAT (green), STS (red) and NAT_STS (orange), as well as a former NAT classification method (black) based on Spang and Remedios (2003), for details see text. Please note the changing maximum counts of observation and corresponding maximum value of the y-axis for individual winters. The coloured numbers in the legend indicate the total number of the corresponding cloud type and the partition on the total number of PSC observations in percent.





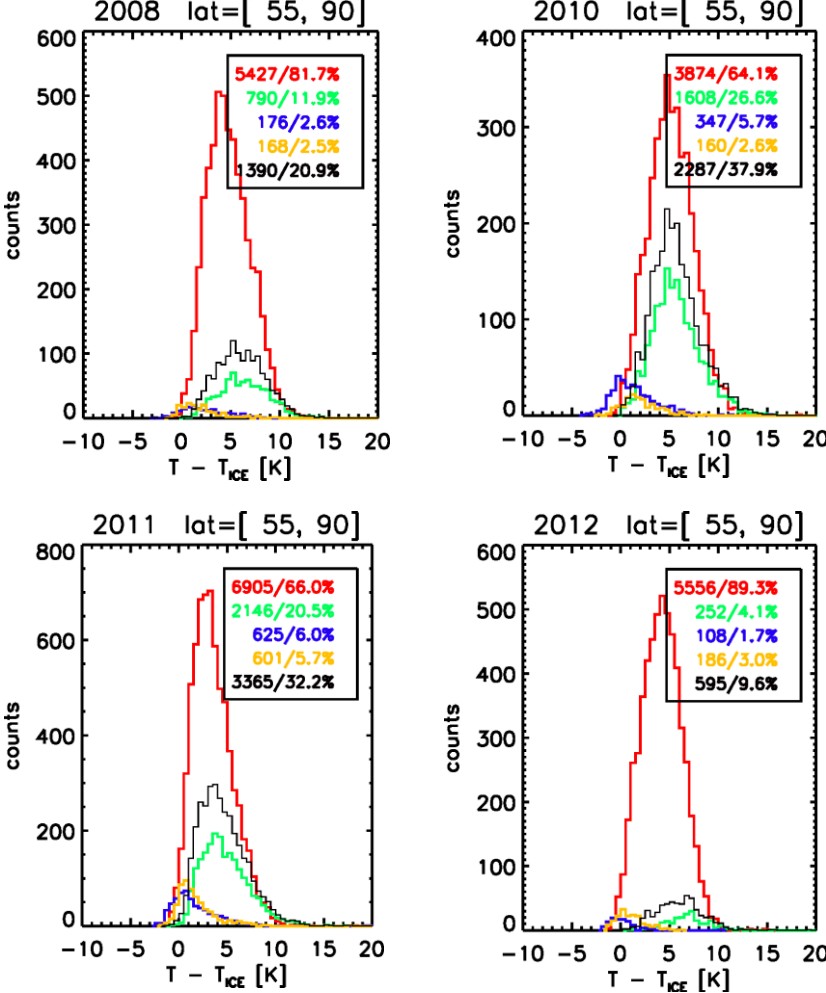

**Figure 4:** Four years of NH polar winter PSC type PDF with respect to $T_{ice}$ (2007/2008, 2009/2010, 2010/2011, and 2011/2012) using the analogue presentation of **Figure 3**. Please, note changing scale of observation counts for each winter.

### 395    3.6   Comparison of retrieved $A_{PSC}$ with CALIOP/CALIPSO

CALIOP classes of PSC types in version 1.0 (https://eosweb.larc.nasa.gov/project/calipso/lidar_l2_polar_stratospheric _cloud_ table) have been compared with the Bayesian classifier results (Version 1.2.8) for MIPAS by Spang et al. (2016). Here, we present a first comparison of the daily and height-resolved area of the hemisphere covered by PSC ($A_{PSC}$, in $10^6$ km$^2$ units), a secondary data product based on the classification results for both instruments where for CALIOP newer Version 2.0

has been used (s. Sec. 2.2). Due to the sparse along track sampling of MIPAS (400-500 km) and the need of an adequate sampling statistics for the computation of $A_{PSC}$, we defined eight equally area-spaced latitude bands from 55° up to the pole,



with 2.3° (250 km) up to 12.2° (1340 km) width in latitude. For each latitude band and 1 km altitude bins for MIPAS respectively 180 m for CALIOP the daily zonal mean occurrence frequency for each PSC class has been calculated, where the mean over the eight bands multiplied by the total area of the polar cap (area >55°N/S = $46 \times 10^6$ km$^2$) is representing a

quantitative measure of the real PSC coverage. This approach bypasses the caveat of the irregular sampling density due to the orbit geometry and avoids artefacts like the overestimated ice partition with respect to the other classes of the Bayesian classifier reported in Spang et al. (2016).

Rex et al. (2004) showed a surprisingly good correlation of the volume of PSCs ($V_{PSC}(T)$) with the degree of Arctic ozone loss. Based on meteorological reanalyses (mainly temperature) $V_{PSC}(T)$ is defined as the stratospheric volume where conditions

are cold enough ($T < T_{NAT}$) for the existence of PSCs and is computed by simple integration of corresponding $A_{PSC}(T)$ over certain altitude layers of the meteorological analyses. Because this quantity is not the real volume or area of PSC coverage the term should be better described with potential volume of PSC formation. However, $V_{PSC}(T)$ is a very useful proxy for ozone destruction ($\Delta O_3$) potential (Rex et al.,2004) and consequently well suited for the validation of CCMs and CTMs over multiple winters and the prediction capabilities of CCMs in a changing climate. Using $V_{PSC}$ as a proxy for polar ozone depletion has

limitations, because even for the same definition for $V_{PSC}$ (which is to some extent arbitrary) the resulting proxy depends on the employed analysis or reanalysis scheme (e.g., Rex et al., 2004, Rieder and Polvani, 2013). This demonstrates that patterns of $V_{PSC}$ derived from PSC observation – put forward here – constitute an important quantity for evaluating the performance of CCMs. Both satellite datasets allow for the first time comparisons of these $\Delta O_3$ proxies based on meteorological analyses with the true atmospheric state of measured $A_{PSC}$ and $V_{PSC}$ (see also **Sec. 4.2**).

For a first impression on the validity of both datasets we compare in **Figure 5** the CALIOP and MIPAS $A_{PSC}$ evolution for the SH winter 2009. The MIPAS measurements needs to be restricted to a maximum latitude of 82°S (82°N for NH winter), the maximum latitude where CALIOP can take measurements. A quantitative comparison is only possible by applying this restriction, otherwise the very high probability to measure ice PSCs in the SH vortex core for mid-winter conditions would bias the statistics due to the lack of CALIOP measurements at these high latitudes (see also Spang et al. 2016). For the

individual cloud classes ($A_{TYPEi}$), STS for MIPAS compared to STS for CALIOP, MIPAS NAT to the sum of NAT mixtures and enhanced mixtures, and MIPAS ice to the CALIOP ice + wave-ice class, we found generally good consistency in the temporal evolution. The NAT class shows in both measurements the largest coverage of the polar cap. The onset of the PSC season 2009 is characterised by first outbreaks of STS detection in mid-May for both datasets, followed by first NAT occurrences between end of May and start of June. All classes show the characteristic descending of the PSC area throughout

the course of the winter. The declining ice probability in September with a slight recovery in mid-September is nicely shown by both instruments.





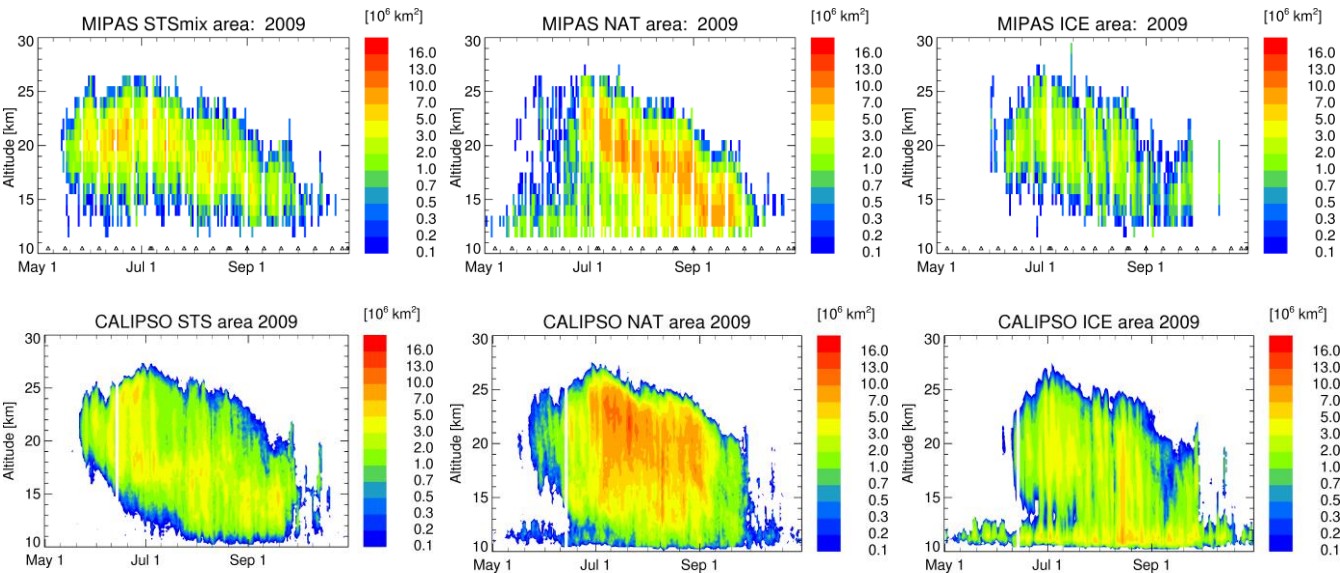

**Figure 5:** PSC type classification comparison between MIPAS (Bayesian classifier Version 1.2.8) upper row and CALIOP (V2) lower row based on the identical computation of $A_{PSC}$ for each dataset. For better comparability MIPAS observations are restricted to latitudes < 82°S, the latitude coverage of CALIOP. Black triangles at 10.5 km altitude in the MIPAS time series are highlighting data gaps, usually caused by the mesosphere/thermosphere measurement mode.

Even the absolute values of $A_{TYPEi}$ for specific days and altitudes show a remarkable good agreement over the entire winter, although some differences are also apparent. For example, the vertical extent all $A_{TYPEi}$ shows systematic offsets on the lower bound due to two effects: (a) The MIPAS analysis is truncated below the 12 km altitude level using a +/- 500 m vertical grid size, and (b) if the analysis is not based on a full radiative transfer retrieval of the measured radiance profile (e.g. an extinction retrieval), than the limb technique has limitations in quantifying accurately the cloud bottom information (Spang et al., 2012). For the presented analysis we used all spectra flagged as cloudy up to 6 km below the actual cloud top height (CTH). In comparison to CALIOP, where the cloud base height is usually well defined in the measurements, the MIPAS restriction shows a tendency to underestimate the PSC bottom altitude and consequently the lower edge in vertical coverage for each PSC type in **Figure 5**. But the CALIOP algorithm is also sensitive to cirrus clouds detected in the tropopause region (~12 km and below). However, in May both sensors show clouds classified as NAT at 12 km and below of similar area. That both methods tend to classify these cirrus cloud detections as NAT clouds need further investigations. This coincidence might be an artefact caused by two completely different reasons, but a real atmospheric effect cannot be ruled out.



## 4 The MIPAS PSC climatology

In the following sections we present details of the new MIPAS PSC climatology. The climatology is the first pole covering, day and night-time dataset that differentiates particle composition. Together with the upcoming CALIOP based PSC composition climatology and pole covering measurements of relevant trace gases for stratospheric ozone chemistry from the

MLS instrument (e.g. Lambert et al., 2012) and from MIPAS (e.g. Arnone et al, 2012), a unique database for polar process studies and climatological aspects of the PSC distribution with respect to ozone destruction is now available under variable winter conditions for both hemispheres.

### 4.1 Overall PSC occurrence

**4.1.1 Hemispheric coverage of mean PSC occurrence**

The climatological mean of all PSC occurrences for nine SH and ten NH winter seasons is presented in **Figure 6**. We limited the analysis to the altitude region 16-24 km where PSC tend to occur, and computed the occurrence frequency by dividing the number of cloudy spectra by the total number of spectra for an equally space of 10° longitude x 5° latitude grid boxes.

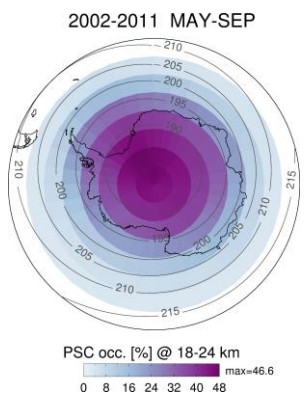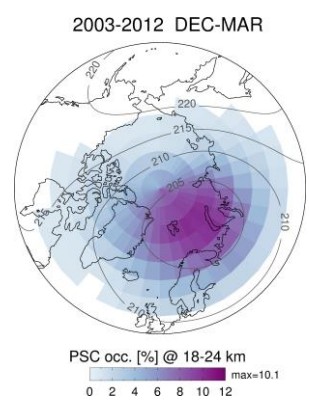

**Figure 6:** Occurrence of PSC spectra with respect to the total number of measured spectra in the complete MIPAS measurement period for SH (left) and NH (right) polar winter conditions. A mean for May-Sep and Dec-Mar for 2002-2011 of ERAi based temperature contours at 50 hPa are superimposed in grey. Please note the different range of the colour bar for both statistics.

Both hemispheres show very different distributions. Compared to the NH, the PSC coverage is significantly larger for the SH and the maximum occurrence frequencies are more than a factor of four higher (~40% versus 10%). In addition the maximum PSC location is centred over the pole for the SH, but is remarkably shifted over the Barents Sea and northern Scandinavia for the NH, which is in line with a shift of the climatological mean temperature contours and minimum in the same region. This shift is caused by stronger planetary wave activity for NH winters with strong wave number 1 amplitudes. Consistently, the





climatological mean temperature in the SH show 10 to 15 K lower mean temperatures than the NH polar vortex. These results
        are in line with the state of the art physical and chemical understanding of polar processes.

        To illustrate the temporal development with respect to the geographical location over the entire winter season **Figure 7** shows
        the climatological mean of PSC occurrence on a monthly basis. The PSC season in the SH comprises May-September, and is
        longer and more intensive than the equivalent December-February season in the NH (March is not presented as the PSC

occurrence is minimal). The SH shows the typically concentric distribution for all months with the PSC occurrence rates
        maximum at or close to the pole and in close correspondence with the temperature contours ($T_{min}$). In the NH PSC occurrence
        rates are more variable from year to year, usually with at most several weeks of severe PSC activity distributed in the December
        to March period. A change of the maximum location from the January to the February mean from the Greenland Sea to the
        east in direction to the north of Russia is mainly caused by individual years with colder stratospheric winters with unusually

large occurrence frequencies and where the geographical location of the planetary wave 1 minimum can be quite different
        from year to year or can even change over the winter.

### 4.1.2 Interannual variability in PSC coverage

        Similar to the altitude resolved $A_{PSC}$ presentation in **Figure 5** it is common for a better characterisation of a specific winter to

compare the temporal evolution of $V_{PSC}$ or $A_{PSC}$ at a single altitude level with a climatological mean of the parameter and its
        variability (e.g. Tilmes et al., 2004). These simple parameters allow also to estimate the ozone destruction potential of a single
        winter (Rex et al., 2004). However, these time series are based on the assimilated temperature distribution from meteorological
        reanalyses, representing the thresholds for PSC existence. However, the actual PSC and PSC-type coverage might be quite
        different.

Here, instead of $V_{PSC}$ we present for MIPAS the maximum area covered by PSC spectra in the altitude range 15 to 30 km. This
        integrated view to the PSC distribution allows us to bypass the difficulties of passive IR limb sounders to retrieve accurate
        cloud bottom information, which is necessary for the derivation of $V_{PSC}$ by measurements. Accurate cloud bottom information
        is especially a problem for optically thicker cloud conditions (e.g. Spang et al., 2012). Nevertheless, this quantity is a good
        estimate of the actual area and overall PSC coverage of the polar vortex, and can also easily be computed from CTM and GCM

model output fields for comparisons and evaluation purposes. **Figure 8** shows the temporal evolution of the integrated
        maximum $A_{PSC}$ ($A_{PSCmax}$) for each winter season (May to October) in the MIPAS measurement period 2002 to 2012 for the
        SH and NH. The winter 2006 showed with maximum values of up to 25 million square kilometres corresponding to more than
        60% of the polar cap south of 55°S the largest cloud area extent. The inter-annual variability is small for the SH compared to
        the NH winters. The absolute values of $A_{PSCmax}$ in the NH are significantly smaller than in the SH (factor 2-3) and periods of

PSC occurrences are restricted to several weeks compared to long extended periods of minimum 4 month (2002) and maximum
        close to 6 month (2006 and 2011) for the SH.

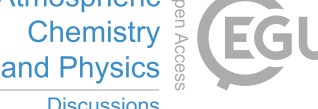



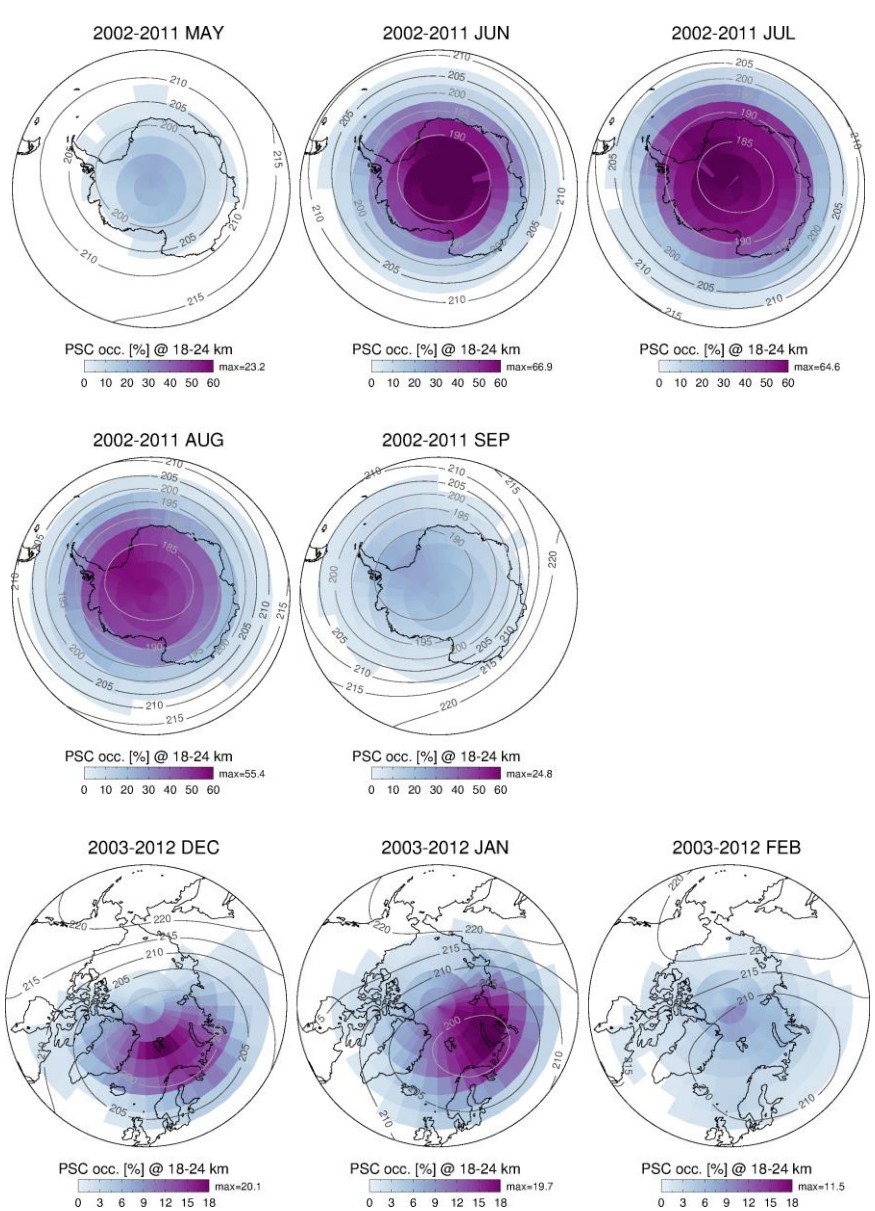


**Figure 7:** Monthly mean of hemispheric PSC occurrence between 18 and 24 km for the months with PSC activity (SH: May, June, July, August, and September; NH: December, January, and February). A mean of 2002-2012 ERAi based temperature contours at 50 hPa are superimposed in grey. Please note the different range in the colour code between SH and NH observations.




The winter of 2002 showed at the end of September an exceptional major warming event in the SH stratosphere, where the
polar vortex split into two vortices and reunited again (e.g. Allen et al., 2003). Temperatures in the stratosphere were higher
than usual during most of this winter, caused by strong planetary wave activity in the troposphere already in May, and several
minor warmings took place during August and September. These very unusual conditions are well reproduced in the PSC
activity measured by MIPAS. Although the measurements started late and are partly interrupted due to instrumental tests
during the commissioning phase of the satellite, $A_{PSCmax}$ for 2002 shows the lowest values of all nine SH winter seasons. A
significant cut in early August (minor warming event, partly masked by a data gap) is followed by an only slight recovery in
PSC activity, with still exceptionally low values compared to other winters, a second breakdown takes place in early
September, and finally the PSCs completely disappear in the second half of September (usually in early October) coincident
with the major warming event starting in mid-September ending with a final warming at 25-26 September 2002 (Butler et al.,
2017). Beside winter 2002 the PSC season 2010 shows remarkably low $A_{PSCmax}$ values between end of July and mid-October,
but no significant indication for a sudden minor warming is for example found in the ERAi (M. Tao, pers. communication).

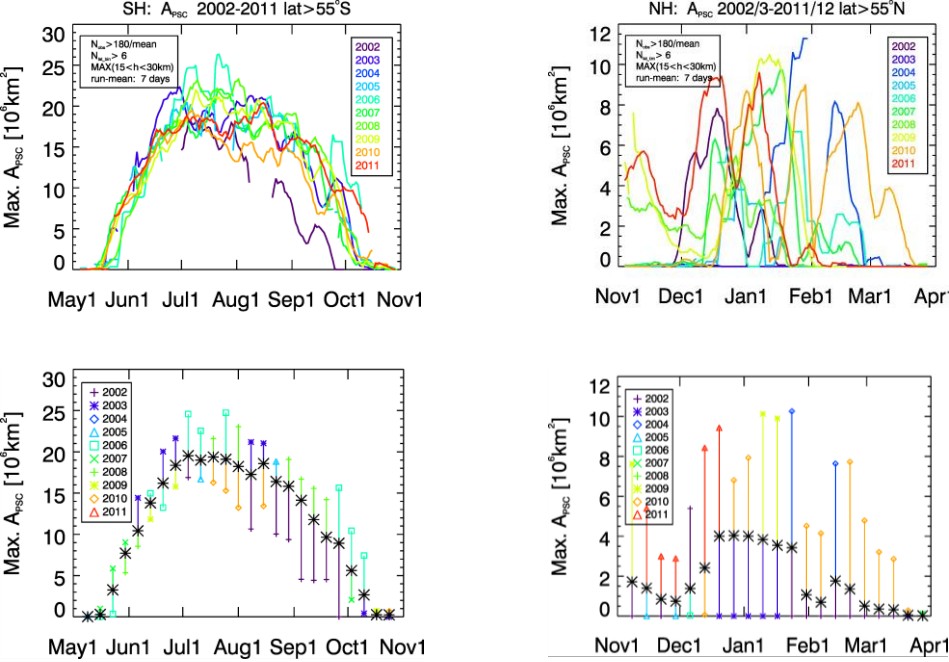

**Figure 8:** Maximum MIPAS area of PSC for SH (May-Oct, left) and NH winter conditions (Nov-Mar, right) computed as
running mean over seven days through all seasons (top). In addition, weekly maximum/minimum (colour coded symbols) and
mean values (* symbol) of $A_{PSC}$ are presented in the bottom row. For both type of representation each season is colour code,
where for the NH the year in the figure legend for example 2002 is linked to the potential PSC season November to March
2002/2003 respectively. For the NH results minimum values of zero appear for several years and the symbols are overlaid.



For the NH observations the winter 2010/11 was exceptional in the sense of unusual three periods of PSC activity from end-December to mid-March. The winter 2011/2012 is highlighted by a very early and extended PSC coverage. The season 2003/4 showed even no PSC activity at all. Special attention has to be taken to the unexpected local maxima in $A_{PSCmax}$ in November

for the years 2008/2009, 2010/2011 and 2011/2012. All these periods are strongly influenced by an enhanced aerosol load of the upper troposphere / lower stratosphere (UTLS) region caused by volcanic eruptions. Dedicated MIPAS aerosol analyses (Griessbach et al., 2016) show that the PSC seasons 2008/2009, 2010/2011, and 2011/2012 are influenced by the volcanic eruptions of Kasatochi (Günther et al. 2017), Sarychev (Wu et al. 2017), Grimsvötn and Nabro (Griessbach et al. 2016, and Günther et al. 2017) eruptions in August 2008, June 2009, May 2011 and July 2011 respectively. Due to the fact that the

Bayesian classifier so far does not include a differentiation between background or volcanic aerosols and PSCs, these events have been treated and classified as PSC particles. The artificial enhanced PSC activity is observed from the lowest altitude level around the tropopause (12 km) up to maximum heights of ~16 km depending on the strength and penetration depth of the corresponding volcanic eruption (see also Sec. 4.2.4).

**4.2 Temporal and spatial analysis of the PSC classes**

In addition to the overall PSC occurrence over the winter, the results of the Bayesian classifier allow a detailed analysis of the horizontal and vertical distribution of individual types of PSCs in the climatological mean or for specific winters.

**4.2.1 Monthly means**

**Figure 9** represents an example for winter 2009 (May-Sep) of the monthly mean occurrence frequencies between 18 and 24 km altitude for the individual classes (STS, NAT, ice, and NAT_STS in each rows) of the classifier. A 10° longitude × 5° latitude grid is used for the statistics.

The different temporal onset of each PSC type becomes here obvious again (see also **Figure 5**). STS dominates the early winter followed by NAT and ice in June/July. The ice is concentrated on the centre of the cold polar vortex surrounded by a belt-like

structure for NAT. This is similar to the plots of individual days presented in **Figure 2** or Fig. 11 in Hoffmann et al. (2017) for the formation of a NAT-belt by MW events downstream of the Antarctic Peninsula. For July (less obvious in June as well) a secondary maximum of NAT occurrence can be seen downstream of the Transatlantic Mountains, an additional hot spot region for MW activity in the Antarctic (Hoffmann et al, 2017). In addition, the June and July occurrence frequencies for ice show an obvious bulge of enhanced activity in the direction of the Antarctic Peninsula, in contrast to the more general concentric

distribution around the pole. If these local maxima are directly linked to MW events a more detailed study is needed, where backward trajectories in conjunction with a microphysical model can establish more profoundly the connection between MW and PSC observations.



**Figure 9:** Monthly polar MIPAS PSC type occurrences between 18 and 24 km for the southern polar winter conditions. Each row shows one month (May to Sep) for 2009 and the respective four main classes STS (orange), NAT(green), Ice (blue), and mixed class NAT_STS (purple colour code) for the Bayesian classifier. Please note the partly different scaling of the colour code for each month and type. In addition, monthly mean ERAi based temperature contours at 50 hPa are superimposed.





Enhanced occurrences for STS are also forming a belt-like structure for June-August. In contrast to the NAT case, this is mainly caused by the synoptic temperatures and also concentric temperature contours in the SH polar vortex, where the quite

cold inner polar vortex (south of 70°S) is dominated by ice particles. This results in a belt-like maximum distribution for the STS occurrence frequency. But in contrast to NAT, STS is not forming a real belt-like structure of clouds on a daily basis of the overall distribution of the PSC classes like in **Figure 2**. The mixed class NAT_STS shows its maximum occurrence in June and in regions with rather low temperatures, where ice may also be formed. This indicates again the very specific occurrence of this class, where potentially all three types of PSCs are involved.


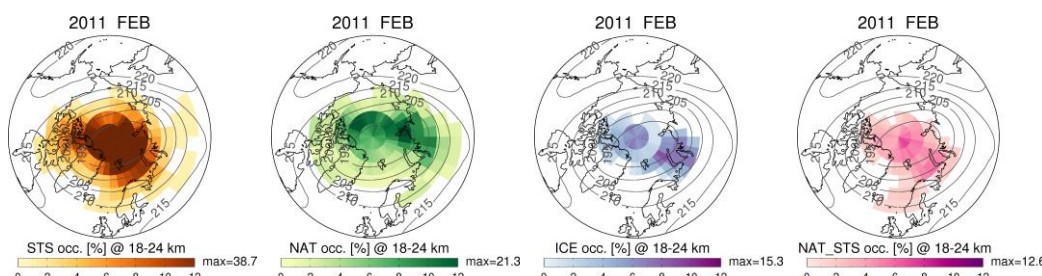

**Figure 10:** Similar representation of the MIPAS Bayesian Classifier PSC classes like in **Figure 9** but for February 2011 only. All PSC classes use the same minimum/maximum values for the colour bar. Maximum values for STS and NAT are significantly larger (38.7 and 21.3 %) than the maximum colour code value (12%).


The NH weekly $A_{PSC}$ distribution in **Figure 8** already showed the high interannual variability with very distinctive and only sporadic PSC activity for a couple of weeks, with sometimes even no activity over the complete winter. **Figure 10** shows only the February 2011 distribution of the four PSC classes of the Bayesian classifier. Due to the generally higher mean and

minimum temperatures of the NH polar vortex no wide spread ice or even no ice at all is observed during typical winter conditions. In this sense winter 2010/2011 was exceptional, in synoptic scale regions with temperatures close and below $T_{ICE}$ spectra have been frequently observed by MIPAS, creating a large area of ice occurrence with maximum values up to 15%. Ice spectra can be found in the NH also in other winters, for example 2009/2010 (von Hobe et al., 2013), but always on a much smaller region (single events) and usually attached to regions where gravity waves may have produced meso-scale temperature

fluctuation sufficient to reach the temperature threshold for ice formation (over Greenland, Scandinavian mountains or the Ural mountains (Hoffmann et al., 2017)).Winter 2010/2011 showed also unusually high ice occurrences in the CALIOP data and an exceptional large ozone depletion over winter and spring close to conditions so far only observed in the SH (Manney et al., 2011, Sinnhuber et al., 2011).



### 4.2.2 Temporal $A_{PSC}$ distributions for each PSC class

**Figure 11** gives an impression of the climatological mean (9 years) daily developments of the area covered by PSCs at a specific altitude (a) and for the specific PSC class of the classifier (b-f). All analysed cloud occurrence frequencies (COF) for the individual PSC classes are transferred to area-weighted information (e.g., $A_{NAT}$, $A_{NAT}$, $A_{ICE}$, and $A_{NAT\_STS}$). The time series for the SH show a number of characteristic and unexpected features:

(a) The maximum in COF for the overall PSC distribution reach values greater than 56% with respect to the polar cap region (latitude > 55°S) which is equivalent to an area size of $15.6 \times 10^6$ km$^2$ or the area of a polar cap completely filled with PSCs south of ~70°S.

(b) There is a significant offset between the mean onset of severe ice, NAT and STS formation during the early SH winter. In a first phase of 2-3 weeks (mid-May to early-June) STS dominates the distribution, followed with the onset of more distinctive ice formation (mid-June), before NAT becomes the dominant PSC type (early-July) for most of the time and altitude range for the rest of the winter. This evolution is in line with the typical mean temperature evolution of the vortex, where temperatures usually fall below $T_{ICE}$ around one month after the PSC onset.

(c) The altitude of the maximum in $A_{TYPEi}$ for each PSC type as well as for the total PSC occurrence is moving continuously downward after mid-July, from around 21 km down to 13 km in direction to the tropopause level. This circumstance follows the typical downward propagation of the vortex cold pool from the mid-stratosphere to the tropopause region for SH polar winter conditions.

(d) The mix-type class NAT_STS shows very similar temporal evolution with height like the STS class, but its major occurrence is limited to the early winter, i.e., the June to early-July period, starting just after ice appears for the first time. This suggests that this class of spectra is dominated by the radiative properties of STS particles under cold conditions, but also with minor up to a comparable contributions in the spectra by NAT particles with larger radii. Particles with larger NAT radii can be deduced only indirectly, because such events show no significant indication of the typical NAT feature at 820 cm$^{-1}$ attributed to small NAT particles ($r < 3$ μm) (Spang et al., 2005a). Consequently, these events might be related to mixtures of STS, large NAT, and potentially a small volume density of ice particles. Following the same arguments, the class *unknown* may have a similar composition. However, these indication of partitioning of PSC classes are very difficult to quantify and would need further detailed investigation with intensive radiative transport modelling of mixed-type clouds.

(e) There is an unexpected early (beginning of May) and surprising low altitude (close to the tropopause region ~12 km) onset of activity for NAT formation in Antarctica. So far it is not completely clear if these events, most likely not related to formation processes of PSCs, are only an artefact of the detection algorithm (e.g. by hypersensitivity) or



related to enhanced aerosol load and/or cirrus cloud formation in the UTLS region. Although this is an interesting detail in the observation, the altitude region is obviously not of primary interest for the PSC research of this study. More detailed analyses are planned for future studies with the database.

The large variability for NH winter conditions, with warm seasons without any PSC sightings and other winters showing

synoptic scaled wide-spread PSC coverage, results in a very large inter-annual variability. Therefore a multi-annual mean like for the SH is not a useful presentation for the NH. **Figure 12** shows an example for the winter 2010/2011, where exceptionally low stratospheric temperatures partly well below $T_{ICE}$ have been observed for several weeks. In addition, these conditions caused an outstanding large ozone loss over the winter (Manney et al., 2011, Sinnhuber et al., 2011).

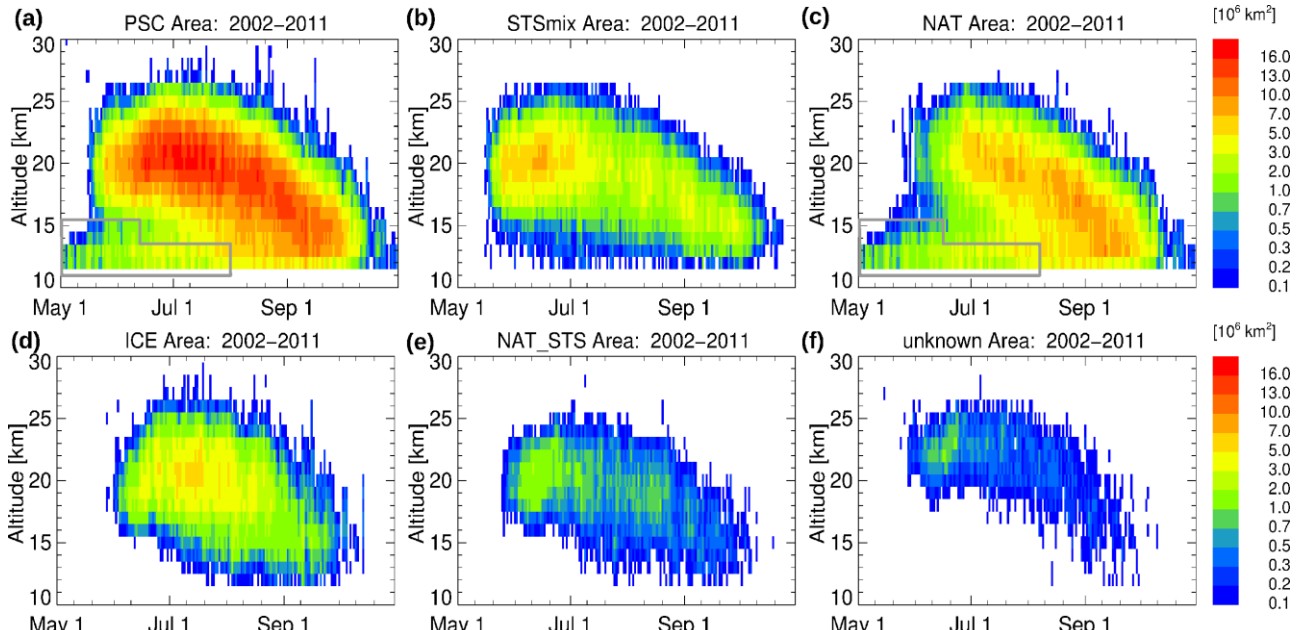

**Figure 11:** Time series of the ten year-mean (2002, 2003 and 2005-2011) of the daily Antarctic PSC occurrence rates weighted with the area of the applied latitude bins south of 55°S in the altitude range 12-30 km. Colour code for (a)-(e) is found at the top right and for (f) at the bottom right side. This approach is equivalent to the partition of areal coverage south of 55°S. An altitude grid of 1 km with 2 km grid size is applied. Presented are all relevant classes of the Bayesian classifier (ice, NAT,

$STS_{mix}$ NAT_STS, and unknown in Fig. (a)-(e)), and finally in Figure (f) for the overall PSC coverage. The grey boxes in the early winter period highlighting regions where polar cirrus at the tropopause and potentially some volcanic aerosols introduce a bias in the PSC detection and classification.

The Arctic winter 2010/2011 shows some characteristic features for the various PSC types. The overall PSC occurrence

frequency and area is already exceptional for a NH winter. From mid-December to mid-March large areas of the polar cap have been covered by PSCs. The peak values reach a PSC coverage comparable to the June conditions in the SH. At end of



December PSCs are formed at exceptionally high altitudes (28-30 km). They have been already described in a previous study as the highest PSCs ever reported in the Arctic (Arnone et al., 2012).

Similar to the SH, STS particles dominate chlorine activation in the early phase of the winter (Solomon, 1999, Drdla and

Müller, 2012). At a later stage NAT detections reach a similar coverage like STS. During three periods ice is reaching an outstanding and significant contribution to the PSC composition compared to other winters (30-50% of all PSC detections for certain days and altitudes). The highest number of the NAT_STS class occurrences are again attached to the coldest time periods and show a very similar pattern like the ice distribution. Finally, the class *unknown* behaves very similar to the SH, and correlates well with the NAT_STS class.


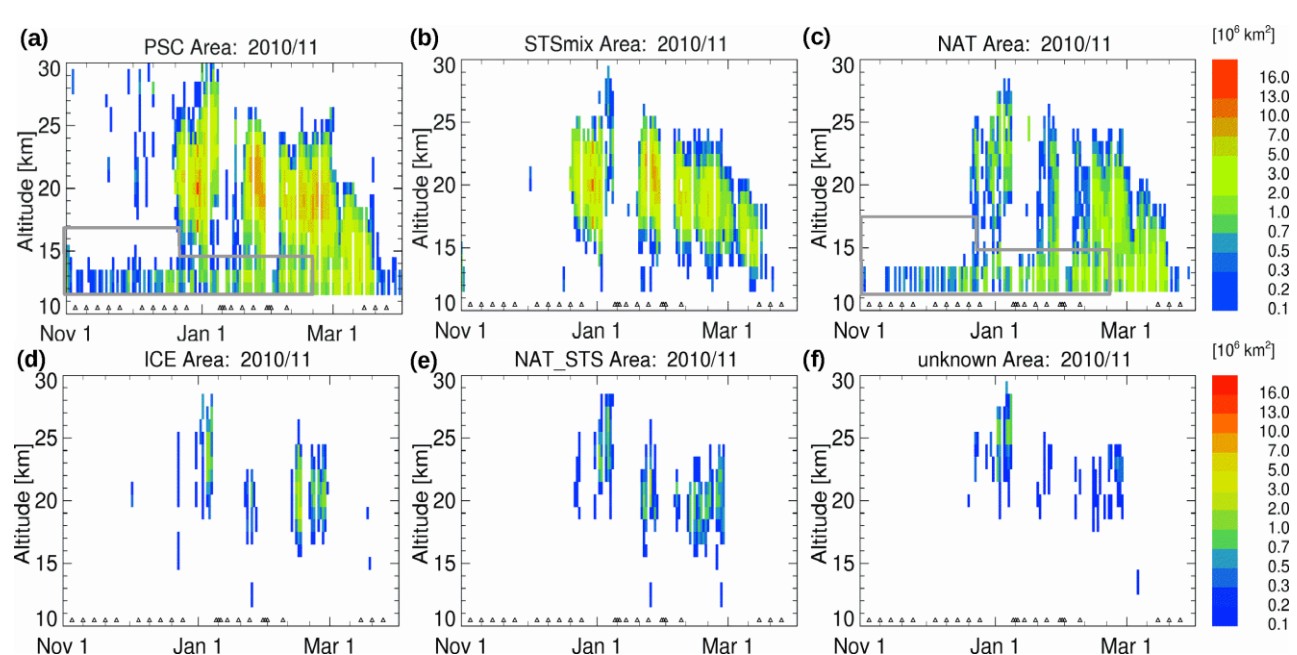

**Figure 12:** Time series of the northern polar winter 2010/2011 of the daily Arctic area PSCs of the applied latitude bins north of 55°N in the altitude range 12-30 km. This approach is equivalent to the partition of areal coverage north of 55°N. An altitude
grid of 1 km with 2 km grid size is applied. Presented are all relevant classes of the Bayesian classifier (ice, NAT, STS$_{mix}$, NAT_STS, and unknown in Fig. (a)-(e)), and finally (e) the overall PSC coverage. The grey boxes are highlighting regions where mainly volcanic aerosols in the UTLS have introduced a bias in the PSC detection and classification. The box is based on all winter (Nov-Mar) observations in the MIPAS measurement period. Triangle symbols at 10.5 km altitude indicate days with no MIPAS mid and lower stratosphere measurements.


As already pointed out in **Sec. 4.1.2** the November-December measurements can be partly contaminated by false PSC detection in the UTLS region (12 km up to 16 km, highlighted by the grey box in Figure **12**). But in contrast to volcanic eruptions in



2008, 2009, and 2011, for late 2010 no severe volcanic plumes have been noticed in the UTLS region of the NH. The early

November to December cloud detection at tropopause level may be most likely related to cirrus clouds similar to the SH cloud detection presented in the multi-annual mean of **Figure 11**.

More detailed statistical analysis for the overall NH observations (not shown) result in the following main findings:

(a) STS is always the dominating PSC type for all observed winter periods. This fact suggests that heterogeneous NAT formation at temperatures higher than $T_{ICE}$ or even higher than $T_{STS}$, like suggested by Hoyle et al. (2013), might not

be the dominating PSC formation mechanism in the NH polar vortex. But large NAT particles may classified as STS by the Bayesian classifier and consequently, it is difficult to draw a distinct conclusion from the MIPAS observations. However, usually the temperatures are not low enough in NH polar vortex for the observation of wide spread ice clouds, but this was the case for winter 2010/2011 (see above) and to some extent also for winter 2009/2010 (not shown).

(b) Ice PSC are a rare observation in the NH polar vortex, but have been observed sporadically with earliest events in the first half of December and the latest by end of February, depending on the individual winter.

(c) The de-nitrification in the mid stratosphere and re-nitrification in the lower stratosphere combined with the subsidence with height of the cold pool over the winter becomes also obvious in the NAT distribution but less pronounced than in the SH.

(d) The maximum altitude of PSC occurrences is higher in the NH than in the SH (1-2 km, and reaches CTHs up to 28-30 km). This difference may be caused by a signal to noise ratio issue in the cloud index detection method for altitudes above ~30 km (Spang et al., 2004, 2012). The cloud index profiles - especially for the cold polar stratospheric night conditions - are starting to get noisy at this level due to the very low radiance values (temperatures) for non-cloudy conditions and the detection of optically very thin PSCs becomes more difficult.

(e) The relatively high NAT occurrence frequency in November in the Arctic in the UTLS region is a prolonged feature and perplexing observation. The phenomenon is definitely related to the enhanced volcanic activity in the time frame of the analysis, where four significant eruptions at mid latitudes had a severe imprint on the UTLS aerosol load in the NH (see also **Sec. 4.1.2**). The aerosols of volcanic origin have similar radiative properties like NAT particles, because no other PSC type shows such a significant occurrence enhancement in the UTLS for the three seasons contaminated

by volcanic eruptions. Why the classifier selects for these altitude for most of the cloud events the PSC class NAT instead of the more sulphuric acid related STS class is still an open question. These events occur mainly in November and at altitudes near the tropopause where the temperatures are clearly too high to form PSCs (both NAT and STS). Consequently, these false PSC detections are easily to exclude from further analysis by simple altitude and temperature thresholds.






### 4.2.3 $A_{PSC}$ based on reanalysis temperature data for the SH winter 2010

Generally, there are difficulties to find a sophisticated approach to retrieve characteristic and comparable parameters from the very different information of remote sensing measurements and the model output of CCMs and CTMs for PSC particles. Some studies use the complex way to retrieve the measurement quantity, e.g. attenuated backscatter signals of lidars or radiances for

passive IR measurements, from the model parameters with a corresponding forward model, for example Engel et al. (2013) for CALIOP or Höpfner et al. (2006b) for MIPAS measurements. For this kind of approach it would be also necessary to sample the model with the asynoptic measurement net of the instrument and for MIPAS to take the effect of limb path integration into account (e.g. Spang et al., 2012). However, this very expensive approach, with large numerical costs and an extensive effort for implementation and validation of the methodology, is only practical for more detailed case studies of a

subset of the data.

**Figure 13** presents the time series of the overall MIPAS PSC and ice class (top row) coverage together with the results of a computation of $A_{PSC}$ and $A_{ICE}$ based on temperature data of a meteorological reanalysis dataset (ERA-Interim). The comparison illustrates the potential benefit of the new climatology for validation and consistency purposes of CTMs and CCMs. A more detailed comparison with the Chemical Lagrangian Model for the Stratosphere (CLaMS) including a sophisticated

microphysical model (Grooß et al., 2014) is presented in a parallel study by Tritscher et al. (2017).

For the ERAi based dataset a grid box is characterised as cloudy if the temperature at the grid point falls below the PSC existence temperature ($T_{NAT}$). If the temperature falls below the existence temperature for ice ($T_{ICE}$) the grid box is additionally marked as an ice cloud. $T_{NAT}$ and $T_{ICE}$ are functions of pressure, $H_2O$ and $HNO_3$ abundance (Hanson and Mauersberger, 1988, and Marti and Mauersberger, 1993). Finally, $A_{PSC}$ and $A_{ICE}$ are computed with an analogous approach as for the MIPAS data

by simply using the grid point of ERAi (longitude, latitude and geopotential height) instead of the tangent height locations of MIPAS. Constant water vapour and nitric acid concentrations are assumed in the stratosphere for the definition of the corresponding existence temperatures of NAT and ice PSCs. For better representation of the dehydration and denitrification process over the winter we computed the distributions for two setups, the high-case (c+d), with high water and nitric acid concentration typical for early winter (4 ppmv and 9 ppbv, respectively), and the low-case (e+f), with mixing ratios more

typical for late winter after severe denitrification and dehydration took place (2 ppmv and 3 ppbv). High and low case together represent a realistic min/max scenario of the $A_{PSC}$ and $A_{ICE}$ distribution based on reanalysis temperatures. More realistic $HNO_3$ and $H_2O$ values could be deduced from for example satellite measurements.



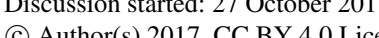

**Figure 13:** Time series of the daily Antarctic area of PSCs ($A_{PSC}$) including all PSC classes and the area of the class ICE only ($A_{ICE}$) for the year 2010 retrieved from MIPAS spectra (Fig. (a) and (b), see also **Figure 11**). This is compared with the standard calculation of PSC areas based on ERA-Interim temperature data (ERAi) with constant stratospheric water vapour and $HNO_3$ values (Fig. (c)-(f)). Results of two setups are presented for $H_2O$ of 4 ppmv and 2 ppmv respectively in (c+d) and (e+f), and for $HNO_3$ with constant 9 ppbv and 3 ppbv respectively (c+e) to illustrate the usually observed effect of denitrification and dehydration in the polar vortex over the winter.

The MIPAS measurements show a significant break in PSC occurrence activity at end of July / early August period, most obvious in $A_{ICE}$ but also the maximum $A_{PSC}$ altitude range (reddish area) is shifting up and down during this phase of the winter. The ERAi analyses show a similar but less pronounced effect, indicating a warming event for parts of the polar vortex





and consequently less PSC occurrence, especially for ice. In addition, the comparison indicates an overestimation of $A_{PSC}$ by the simple temperature based estimate of these quantities for most of the winter. Larger $A_{PSC}$ values over an extended altitude range are predicted even in the ERAi low-case. For $A_{ICE}$ a mixture of the ERAi high and low-case could possibly fit the vertical distribution quite well, but the high-case with a realistic value of 4 ppmv for early to mid-winter conditions is creating a by far too large $A_{ICE}$ in the ERAi analysis compared to the MIPAS measurements. Potential reason for this discrepancy could be:

(a) a potential cold bias in the reanalysis data. This would be in contradiction to a very recent study of reanalysis temperature data which shows a slight warm bias for the polar lower stratosphere for ERAi (Lambert and Santee, 2017).

    (b) a lack of sensitivity of MIPAS to detect all forms of ice clouds. For example optically very thin clouds or clouds only partly in the relatively broad vertical FOV of MIPAS (3-4 km) might be missed or misclassified. However, this
explanation seems unlikely due to the extremely high detection sensitivity of an IR limb sounder with respect to ice clouds and usually strong cloud signals even when ice is covering only a part of the measurement volume (Spang et al., 2012, 2015, 2016).

    (c) the simple temperature-based method is not accurate enough, to describe the occurrence of PSCs.

The temperature based $A_{PSC}$ can only be a rough proxy for the real PSC coverage because detailed microphysical modelling
needs to be considered for an adequate estimate of the PSC type coverage. Studies by Engel et al. (2013) and Grooß et al. (2014) showed that detailed microphysical modelling of PSCs is essential to get an adequate agreement in the PSC coverage and types distributions along segments of CALIOP orbits covered by PSCs.

Nevertheless the comparison shows that SH PSC occurrence can show a substantial interannual variability in the temporal evolution of the PSC type occurrence for particular winters. The major stratospheric warming event in 2002 with a split of the
vortex and the ozone hole has been reported already above and is highlighted in the literature as a unique event (e.g. Newman and Nash, 2005). But also the winter 2010 shows an enhanced variability in the PSC distribution that is related to a less pronounced warming event in the stratosphere.

### 4.2.4 Interseasonal time series of PSC and ice coverage

The results of **Sec. 4.2.3** and **Sec. 4.1.2** already show the benefit a climatology of PSC types can provide for validation purposes of models and for complementary measurements (ground based, air and space borne sensors) as well as reanalyses. **Figure 14** and **Figure 15** presents a comparison of the complete time series of $A_{PSCmax}$ and $A_{ICEmax}$ retrieved from the MIPAS PSC climatology with the ERAi based maximum $T_{NAT}/T_{ICE}$ area for the high/low-case scenario of $H_2O/HNO_3$ concentrations (see Sec. 4.2.3). The MIPAS time series are smoothed by taking a seven day running mean. Data gaps in the time series due to
changing measurement modes and restricted operation times of the instrument in 2005 and 2006 makes this smoothing essential to obtain a continuous time series.



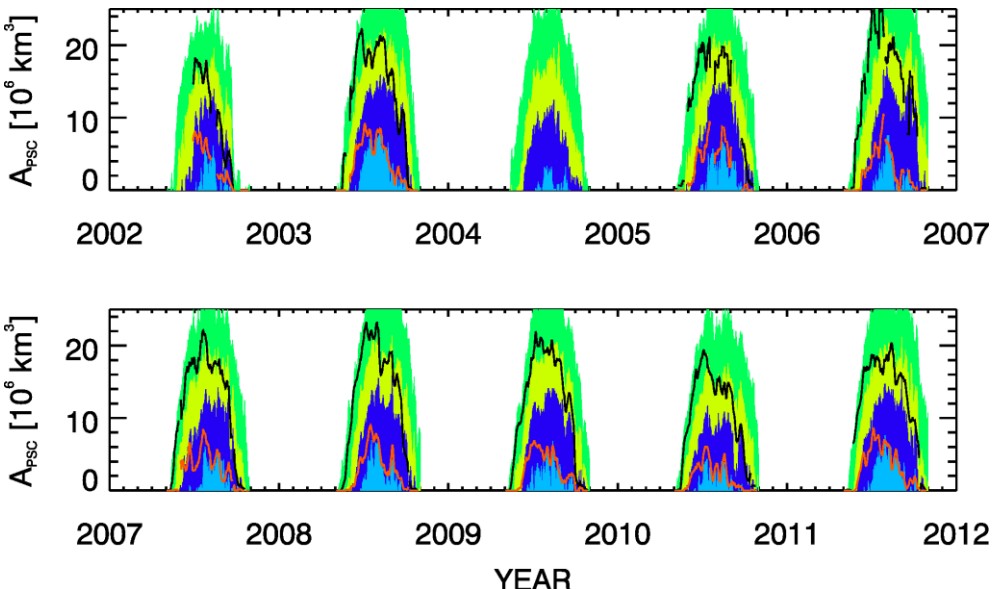

**Figure 14:** SH inter-seasonal variation in $A_{PSCmax}$ between 15 and 30 km with black line for MIPAS (7 day running mean), green and light-green coloured areas retrieved from ERA-Interim daily temperature fields and fixed $H_2O/HNO_3$ profile for high and low-case (s. **Sec. 4.2.3**) and $A_{ICE}$ with a red line for MIPAS, blue and light-blue bars for ERA-Interim high and low-case respectively. In the Antarctic polar winter 2004 MIPAS measurements are not available.

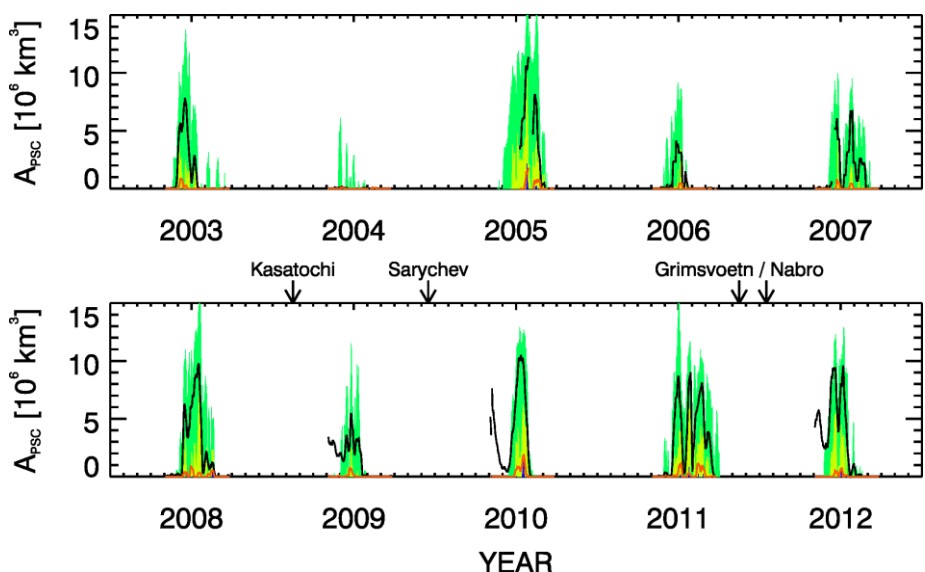

**Figure 15:** NH inter-seasonal variation of $A_{PSC}$ and $A_{ICE}$ based on ERA-Interim and MIPAS (see **Figure 14**). In the lower panel dates of NH volcanic eruptions with stratospheric aerosol burden are superimposed to highlight the link to the artificial $A_{PSC}$ enhancements in November 2008, 2009, and 2012 and which are not caused by PSCs. Please note that data before November 1 are not considered in the analysis.



The comparison shows a number of characteristic features:

(a) The onset of PSC formation is remarkably well reproduced for the more realistic ERAi high-case. Nearly all winter seasons in the SH show small offsets in the range of a couple of days.

(b) The NH shows larger differences for the onset, indicating difficulties in the prediction of PSC occurrence by the simple methodology for the more variable conditions of NH polar stratosphere.

(c) Even the SH onsets for ice formation agrees reasonably well for the more realistic ERAi high-case.

(d) The end of the PSC seasons, which should be better compared to the ERAi low-case, show a less good correspondence (earlier interruption of PSC activity in the observations). Re-nitrification and rehydration at lower altitudes will modify the trace gas distributions but this is not considered in the simple high/low-case approach.

(e) The absolute values of $A_{PSCmax}$ and $A_{ICEmax}$ compare remarkably well for the low-case scenario, whereas the high-case shows overestimation for the SH and NH conditions. More reliable information on the spatial and temporal evolution

of $HNO_3$ and $H_2O$ over the winter for example from satellite measurements would certainly improve absolute values of the ERAi temperature based parameters. CTM and CCM deliver simultaneous and consistent trace gas and temperature information. Comparison with the MIPAS time series might help to improve and consolidate the PSC parameters of the model.

(f) Volcanic eruptions have produced artificial PSC coverage in the NH for November 2008, 2009 and 2011 by severe

aerosol load in the UTLS (see also **Sec. 4.1.2**). Especially, the eruption of Sarychev in June 2008 created a long lasting aerosol cloud covering large parts of the NH up to the pole (Wu et al., 2017).

The results of the comparison between MIPAS PSC observations and simple temperature based PSC proxies show that the overall winter evolution of PSCs can be modelled reasonably well by the simple temperature based estimates of the PSC proxies. A similar approach applied to the output parameters of global models could be a valuable tool to quantify the quality

of PSC related processes in CCMs and GCMs.

Comparisons of the hemispheric distribution of PSC classes on a daily basis like in Figure 2 show frequently discrepancies with respect to the superimposed contours of temperature thresholds although the comparison of the overall integrated information of $A_{PSCmax}$ or $A_{ICEmax}$ is looking promising. The discrepancies are highlighting the difficulties to predict the detailed temporal and spatial distribution of the PSC types correctly, in situations when additional and more detailed modelling of the

microphysical and dynamical processes (e.g. heterogeneous/homogenous nucleation, mountain waves) is becoming essential.

## 5  Summary and Conclusions

The unique dataset of ten years of MIPAS limb IR spectra has been analysed with a Bayesian classifier for the differentiation of various PSC classes (Spang et al., 2016). We detected cloudy and classified spectra of ten NH and nine SH polar winters.



The classification delivers the main classes STS, NAT and ice, but also less stringent classifiable types of spectra described with NAT_STS for a mixed-type class and spectra which cannot reliably classified at all (so-called *unknown* class).

The classification is guided by the characteristic spectral shape in the emission spectra of the different PSC particles and is sensitive to specific features or characteristic gradients over the selected atmospheric window regions of the IR spectrum. The measured radiation is frequently dominated by the spectral shape of only one particle type but in the case of mixtures this type is not necessarily the particle type with the largest volume density or area density along the line of sight of the instrument. Due to the long limb path of the instrument the detection sensitivity is excellent and in the range of spaceborne lidars. The temporal and vertical evolution of area covered by all PSC classes ($A_{PSC}$) and specifically ice ($A_{ICE}$) shows a good correspondence with the new Version 2 dataset of PSC detection and classification from the CALIOP lidar measurements onboard the CALIPSO satellite.

Various comparisons and applications of the MIPAS climatology have been presented in this study and the main results will be highlighted as follows:

1) PDFs of $T$-$T_{ICE}$ for the main PSC classes of the Bayesian classifier using ERAi coincident temperatures show reliable distributions with respect to the current understanding of PSC particle formation. The mixed-type class NAT_STS shows a striking tendency for quite cold temperatures and is mainly observed in the first half of the SH winter. This may point to a special mixture of NAT and STS with some potentially weak contributions of ice.

2) The monthly horizontal distribution in the PSC class occurrence frequencies show pronounced geographical regions with enhanced PSC activity for the SH. Some of these regions show for specific months of the multi-annual mean a link with regions well known for strong MW activity (e.g. a belt downstream of the Antarctic Peninsula with enhanced NAT occurrence, and a local maximum in the ice occurrence over AP). Ice is usually dominating the inner part of the polar vortex and most for July and August consistent with the development of the stratospheric mean temperature distribution.

3) $A_{PSC}$ (and $A_{TYPEi}$) retrieved from the classifier results can be used as a quantitative measure for the daily evolution and strength of the PSC distribution over winter. The parameters are useful for comparison with model-based or simple analysed temperature-based $A_{TYPEi}$ time series for validation purposes. In a second step, the conclusions may allow the partially simple microphysical schemes applied in CCM and CTMs to be improved and optimised.

4) A comparison of the MIPAS-based and simple temperature-based $A_{PSC}$ and $A_{TYPEi}$ distribution shows, that models with a simple temperature threshold parameterisation of the PSC coverage require a good representation of the temporal and spatial water vapour and nitric acid distribution over the winter to adequately reproduce the 'real' measured $A_{TYPEi}$ distribution.

The MIPAS PSC climatology (2002-2012) and the 11 year time series of CALIOP measurements (2006-present) constitute an unprecedented and comprehensive overview of the formation processes and temporal development of PSCs in the NH and SH




winter polar vortices on vortex-wide scales. Together with the MLS trace gas measurements on EOS-Aura process studies can now address a number of still imprecisely understood processes (formation of large NAT particles, heterogeneous and/or homogeneous nucleation, types of condensation nuclei fostering PSC formation). Moreover, combining the PSC information

from MIPAS and CALIOP with the measurements of key chlorine containing species from MLS and MIPAS will furthermore allow the impact of different PSC types on the chlorine chemistry to be evaluated and compared to model results.

These improvements will contribute to a better and more reliable prediction of the future polar ozone in a changing climate. Upcoming results of the currently wide-spread investigations for improvements and more realistic handling of the PSC schemes in state of the art CTM and CCMs will need comprehensive datasets for consistency tests and validation purposes.

Currently, there is no comparable combination of satellite instruments part of the strategic programs of the international space agencies. Consequently, these datasets will have to act as a reference climatology for the hemispheric PSC coverage and type distribution for a very long time.

## 6  Datasets and access to the data

The ERA-Interim dataset has been made available for download by ECMWF (Dee et al., 2011). The meteorological data like temperature, pressure and geopotential height are interpolated to the measurement location of MIPAS (tangent point), and are part of the climatology data product.

The MIPAS PSC climatology consists all measurement points of the mid-stratosphere down to the lowest tangent point of a single measurement profile for the time period June 2002 to Mar 2012. The dataset is available in user-friendly NetCDF format

(https://www.unidata.ucar.edu/software/netcdf). In addition to the geospatial information (altitude, longitude, latitude) the parameter *class* includes the classifications result [-1, … , 7] for the classes [no-cloud, unknown, Ice, NAT, STS, ICE_NAT, NAT_STS, ICE_STS]. In addition, the corresponding three product probabilities $P_{ICE}$, $P_{NAT}$ and $P_{STS}$ used in Bayesian classifier are listed for each cloudy spectrum and can be used as a quality check of the classification approach.

The dataset is registered by re3repository.org (**http://www.re3data.org/repository/r3d100012449**) with a corresponding

digital object identifier (http://doi.org/10.17616/R3BN26) (r3edata.org, 2017) and is available for download for the scientific community via **http://datapub.fz-juelich.de/slcs/mipas/psc** (daily files for May-October for the SH and November-March for the NH PSC seasons). In addition, preview images are available on a daily basis and at a number of potential temperature intervals. The presented secondary data products like $A_{PSC}$ or PSC occurrence frequencies can be drawn on request from the first author.


## 7  Acknowledgement

We gratefully acknowledge ECMWF, ESA, and NASA for the preparation of the ERA-Interim dataset, the MIPAS Level 1 data, and the CALIOP PSC data, respectively. R.S. thanks S. Griessbach (FZJ) for discussions and references on the topic of





the volcanic aerosol load of the stratosphere. M. Tao (FZJ) has kindly provided the characterisation of minor/major warming
events for the southern hemisphere. Finally, the authors like to thank the International Space Science Institute (ISSI) and
Stratosphere-troposphere Processes And their Role in Climate (SPARC) for supporting the activities on the compilation of a
PSC climatological dataset.

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
