# Peer review of "A climatology of polar stratospheric cloud composition between 2002 and 2012 based on MIPAS/Envisat observations"

_Atmospheric Chemistry and Physics, 2017_

## Referee Comment (RC1) · E. Remsberg (Referee) · 28 Nov 2017

General comments: This manuscript is a comprehensive summary of findings from ten years of MIPAS data on PSC area and composition and for separate northern and southern winters. I commend the authors on their choice of figures and for their concluding remarks about each one throughout the text. The channel placement on MIPAS is, perhaps fortuitously, well suited for teasing out composition and/or PSC type. It is very likely that this will be the only satellite climatology on PSC composition for years to come. They also point to earlier publications that give more details about their methods for identifying PSC type. It would be helpful to have an additional paragraph

and references about uncertainties for this climatology.

After reading the excellent introduction section of the manuscript, I was hoping for some guidance in Section 5 on how to use the present climatology for predictions of future ozone loss from chemistry-climate models. The authors are not consistent in this regard. For example, at line 803 they say "that the overall winter evolution of PSCs can be modelled reasonably well by the simple temperature based estimates of the PSC proxies". But, earlier on line 753 they conclude that "the simple temperature-based method is not accurate enough, to describe the occurrence of PSCs". At line 413 they say that "VPSC(T) is a very useful proxy for ozone destruction potential (Rex et al., 2004). . .", but they then follow at line 714 by implying that a more detailed comparison with CLaMs including a sophisticated microphysical model (Grooss et al., 2014) was necessary as presented in a parallel study by Tritscher et al. (2017).

So how should a modeler proceed? Does one need to employ GCM output or as-similate data on temperature, nitric acid, water vapor, and winds (at a minimum) to determine threshold conditions for PSC type, and then use that information to calculate ozone loss potential for future years? Are you recommending that the models include a parameterization for PSC occurrences and types and that one should validate its calculated PSC distributions against the MIPAS findings for a specific year and winter before going ahead with multi-year predictions of ozone loss? Or is it still to be determined how best to employ this climatology for such studies?

Specific comments: I find that the PSC class PDF distributions as a function of T-Tice ought to be most useful (Figures 3 and 4), although you do not indicate what is the related atmospheric pressure. Figure 2a of Pitts et al. (2013) shows that the threshold T for formation of STS versus NAT depends on pressure, as well. Further, at line 370 you say that "a similar temperature analysis of PSC classes by Pitts et al. (2013, Fig. 8 and Fig. 9) . . . shows comparable results with respect to the PDF maxima location and shift between the PSC types" (in your Figures 3 and 4). However, I find that T-Tice for the center of the NH PDF of STS in their Fig. 8 is more like 1 K, while in your Fig. 4

it is nearer to 5 K. Those two results do not agree, in my opinion!

I am familiar with trying to identify PSCs from a precursor dataset to that of MIPAS, i.e., from the Nimbus 7 LIMS experiment. For the LIMS profiles I was able to determine the occurrence of PSC-like emission signatures as a function of threshold and/or existence temperature, but not its composition or type (see Remsberg and Harvey, Atmos. Meas. Tech., 2016). Your Figure 4 indicates that the predominant PSC type for NH winter is STS with its temperature threshold of ∼192 K. I found a similar threshold temperature for PSC occurrences from LIMS, but could not verify that it was most likely STS based on Pitts et al. (2013, Fig. 8). Information in the literature at that time about likely composition was not as precise as you imply, in my opinion, and may still not be. Some guidance on this point would be welcome.

———————————————————

---

## Referee Comment (RC2) · Anonymous Referee #2 · 21 Dec 2017

Review of 'A climatology of polar stratospheric cloud composition between 2002 and 2012 based on MIPAS/Envisat observations', by R. Spang et al., submitted to ACPD

Summary:

Spang et al. have a thorough, interesting, well-written and novel manuscript describing a PSC climatology constructed from MIPAS satellite observations. Spang et al. use a recently-developed Bayesian classifier of MIPAS data to determine the composition of Polar Stratospheric Clouds (PSCs) using the decade of available MIPAS observations. The authors confirm the classification scheme's validity through comparisons with CALIOP data and the probability of each PSC class relative to the ice formation

threshold is consistent with our understanding. We see, through results presented in this paper, the PSC occurrences (importantly, with MIPAS observations stretching to the pole) in each hemisphere along with PSC areas each year – the latter linking in with ERA-Interim reanalysis fields as a pointer as to how GCMs and CCMs might evaluate their own performance.

I recommend its publication in ACP following their addressing of the following minor comments.

Minor Comments:

Line 59: I guess not only in the Arctic this is important. . .

Paragraph starting line 83: Your statement '. . .are so far only very limited. . .' when discussing the current (prior to this MIPAS paper) state of PSC type observations. I would argue strongly that the previously-derived CALIOP satellite PSC climatologies render this statement incorrect. See papers by Pitts et al. (ACP, 2007, 2009 etc). I recommend you be up-front about the previous CALIOP satellite climatologies and note its achievements and strengths, but then state clearly where your new MIPAS dataset fills in gaps left by CALIOP, and the MIPAS dataset strengths. This could be done in a new paragraph following line 110.

Line 342: Provide a reference that SH winter 2010 was a warm winter. (de Laat & van Weele, Scientific Reports, 2011 , doi:10.1038/srep00038)

Figure 5: MIPAS NAT in 2009 (middle top figure) shows NAT between mid May and mid June below about 17km altitude. This is not seen in the CALIOP NAT area (middle bottom figure). This is separate from the tropopause-level NAT &ice issues which you discuss in the text. Can you explain why MIPAS sees NAT here but CALIOP doesn't?

Line 529-530: You should refer to de Laat & van Weele (Scientific Reports, 2011), doi:10.1038/srep00038 who show anomalously warm temperatures in MLS data and discuss the role of minor SSWs in 2010. Remove and replace line 530 and the personal

communication with reference to this paper instead.

Technical Corrections:

I suggest a careful re-read of the manuscript to avoid spelling, sentence structure, and grammar errors. I picked up the following:

Line 74: '... type of PSCs are present...'

Line 103: Reword, it's unclear

Line 147: 'optically thin'

Line 177: 'most sensitive (Spang)'

Line 232: Replace 'whereby' with 'on the other hand,' or similar phrase

Figure 1 caption 'colour ratios'

Line 287: Reword the sentence beginning 'Afterwards a very...' – not clear.

Line 346: again, replace 'whereby' in this case with 'whereas'

Line 350: do you mean 'analogous'?

Line 372: seems you have cut out something here. What is 'u, the CALIOP analysis'?

Figure 4: Suggest for NH you write 2007/2008 etc. in the titles because (a) it is clearer and readers will know instantly which winter you mean; (b) that's what you have in the caption

Line 403: Unclear sentence. Reword

Line 615: 'spectra is not only dominated...'

Figure 13: Make the dates the same between these figures. The top two have 'May 1, Jul 1...', the rest have '01.05.10, 01.07.10...'

Figure 14 & 15 captions: Shouldn't this be A_{ICEmax} not, as currently written,

$A_{ICE}$?

Line 799: '2011 by enhancing the aerosol...'

Line 801: 'classifier to distinguish various PSC...'

---

## Referee Comment (RC3) · Anonymous Referee #3 · 22 Dec 2017

The paper report an extensive presentation of PSC climatology over both poles. PSC have been classified into seven classes: those formed by the three canonical PSC particle types (NAT, STS, ICE) as well as external mixtures of them, and unspecified clouds. The classification algorithm is discussed clearly but briefly, as more detailed descriptions are already present in the literature. Results of PSC classification are then compared with the satellite borne CALIOP lidar classification, used as a benchmark, in term of PSC occurrence vs altitude and time of the year, for a particular year, over Antarctica. The comparison result is quite satisfactory and put trust in the results of the MIPAS classification that are discussed further on. PCS climatology is then described in term of pdf of PSC occurrences vs temperature, and studied in terms of their

geographical distribution, altitudes, and temporal evolution. This is extensively done for the whole PSC dataset spanning several years, over Antarctica, while the author have sometimes restricted themselves to study a single year over the Arctic, due to the larger variability, and hence scarce representativeness of quantities averaged over the whole dataset. Observational climatologies are also compared with PSC occurrences estimated by a simple temperature-threshold based approach on ERA-i meteorological fields. The paper present a unique dataset of paramount importance, is clearly written and well detailed and referenced, surely deserves publication. However the author should correct some inaccuracies in the text, and may consider some suggestion for improving its readability; as instance, the paper is quite lengthy and the authors may consider (this is not a mandatory request) to shift some material in a supplement. These, and other minor issues, are detailed hereafter:

(43) "PSC... and are formed of particles that are classified into three types..." as the particles are of three types, but coexist, giving rise to PSC classes more than three.

(56) maybe her is "typology" instead of (or in addition to) "concentration"

(60) Here, for completeness, the authors may also wish to quote the extensive dehydration that PSCs can induce in Antarctica.

(80) "…faster…" unclear. Faster than what?

(169) "… with altitude, and with altitude…"

(191) "600000 modelled spectra with varying PSC types:" here it is unclear to me whether this forward model account for external mixtures of the three PSC particles. It seems otherwise, also in view of the following discussion, but maybe this this should be made clearer here.

(372) Typo.

(439) Typo.

(442) Typo.

(440-450) The comparison shows good agreement, but I had some difficulty following the reasoning that tries to explain the only evident discrepancies between the two datasets: NAT and ICE in the low altitudes at the beginning of the season. While the absence of ICE at low levels in MIPAS may be due to the vertical truncation of its data, and while the detection of NAT around 12 km is commented for both datasets, the increased presence of NAT in the MIPAS measures between 12 and 15 km seems to me not well addressed. The authors could similarly comment on that small discrepancy.

(542) Unfortunately in fig. 8 the years 2007 and 2008 are displayed with shades of green too close for me to be able to distinguish them.

(543 and following) Here, and elsewhere, the authors focus their attention on a particular Arctic winter, rather than reporting the climatology of mean values. I do not agree with this choice, given that the particular winter is, as highlighted in the text, exceptional, and therefore not very representative. It is also true that, given the high arctic variability, even the average values are not very significant, but this can be highlighted, thus commenting on the low representativeness of the average conditions, while reporting the climatology over the whole dataset and transferring the discussion on the specific winter 2011 in a supplement.

(612) Here the author may also quote that, to a lesser extent, also the denitrification and dehydration play a role in the downward propagation of PSC occurrence.

(631) See comment for (543 and following).

(685-689) This interesting difference between Arctic and Antarctic seem to be explained by an artifact, but I did not get the explanation in full. Is this difference arising because high, thin arctic clouds are warmer that the corresponding Antarctic ones, at the same altitude and with the same optical thickness? If so, the authors may consider to rephrase the paragraph to make such statement clearer.

(690-700) Interesting feature. Have the authors tried to apply their classification algorithm in non-polar, volcanically contaminated stratospheric regions and see whether there too, the algorithm recognizes presence of NAT?

(701-762) The author may consider to shift the whole paragraph in a supplement, and to quote only the main result in the manuscript.

(711) ""... overall MIPAS PSC..." all? Antarctic?

(795) "... would certainly..." I would use "... could..." as you don't know unless you try...

(803) this statement seem in conflict with an earlier one at (753). Maybe one of the two should be rephrased.

---

## Author Comment (AC1) · 9 Mar 2018

**Reply to reviewer E. Remsberg:  (reviewer #1)**

We thank the reviewer for the helpful comments and suggestions on the manuscript. Please find below the point-by-point response and the changes in the manuscript. Replies are presented using times roman fonts. New or reworded text passages in the revised version are highlighted in *italic*.

Reviewer Comments:

The authors are not consistent in this regard. For example, at line 803 they say "that the overall winter evolution of PSCs can be modelled reasonably well by the simple temperature based estimates of the PSC proxies". But, earlier on line 753 they conclude that "the simple temperature-based method is not accurate enough, to describe the occurrence of PSCs". At line 413 they say that "$V_{PSC}(T)$ is a very useful proxy for ozone destruction potential (Rex et al., 2004). . .", but they then follow at line 714 by implying that a more detailed comparison with CLaMs including a sophisticated microphysical model (Grooss et al., 2014) was necessary as presented in a parallel study by Tritscher et al. (2017).

We revised those potentially misleading and contradictory statements. We had discussed the advantages and limitations of the $V_{PSC}$ proxy already in the original manuscript (now lines 416-424) but also added an rephrased in Section 4.2.3

*(c) the simple temperature-based method is not accurate enough to describe the occurrence of PSCs with respect to vertical distribution and temporal evolution over the winter.*

And at the end of Section 4.2.4:

*The results of the comparison between the temporally smoothed and vertically integrated maximum area of PSC coverage of MIPAS with the simple temperature based PSC proxies show that the overall winter evolution can be modelled reasonably well. Although, more detailed and less smoothed analyses for individual winters of $A_{TYPE}$ show significant differences (see Figure 13). However, a similar approach applied to the output parameters of global models could be a valuable tool to quantify the quality of PSC related processes in CCMs and GCMs.*

So how should a modeler proceed? Does one need to employ GCM output or assimilate data on temperature, nitric acid, water vapor, and winds (at a minimum) to determine threshold conditions for PSC type, and then use that information to calculate ozone loss potential for future years? Are you recommending that the models include a parameterization for PSC occurrences and types and that one should validate its calculated PSC distributions against the MIPAS findings for a specific year and winter before going ahead with multi-year predictions of ozone loss? Or is it still to be determined how best to employ this climatology for such studies?

The best approach to compare our PSC climatology with mode results still needs to be determined. It is also not clear, if a single approach serves all needs. The approach depends strongly on the model to be validated, i.e. how detailed the model treats the PSC processes and which parameters are to be 'validated' (e.g. overall column ozone depletion, ozone trends, PSC coverage or volume). For more sophisticated PSC schemes in CCMs, which are taking the formation of PSC types into account, it is meaningful to compare detailed PSC type distributions. We noted, that this is done in the parallel study (Tritscher et al., 2018). In our study, we only like to show the potential of the new database to improve and validate CCMs or CTMs. Further studies should explore and develop detailed procedures on the specific scientific question to be addressed.

Specific comments:

1) I find that the PSC class PDF distributions as a function of T-Tice ought to be most useful (Figures 3 and 4), although you do not indicate what is the related atmospheric pressure. Figure 2a of Pitts et al. (2013) shows that the threshold T for formation of STS versus NAT depends on pressure, as well.

The pressure dependence is already considered by applying T-$T_{ICE}$ instead of T in the PDF distributions. This approach is frequently used in PSC studies and is also illustrated in Figure 2b of Pitts et al. (2013).

2) Further, at line 370 you say that "a similar temperature analysis of PSC classes by Pitts et al. (2013, Fig.8 and Fig. 9) . . . shows comparable results with respect to the PDF maxima location and shift between the PSC types"

(in your Figures 3 and 4). However, I find that T-Tice for the center of the NH PDF of STS in their Fig. 8 is more like 1 K, while in your Fig. 4 it is nearer to 5 K. Those two results do not agree, in my opinion!

This is a justified objection. Both results for STS do not seem to agree very well for the maximum location. We refined this section by highlighting the discrepancy. In addition we deleted the sentence:"This is an independent endorsement in the reliability of the new MIPAS approach."

Because there is so far no explanation for this difference (see following paragraph), we avoid to present details on speculative options and just mentioned the larger difference in maximum location of the PDF between ice and the other types in Figure 4:

*A similar temperature analysis of the PSC classes by Pitts et al. (2013, Fig. 8 and Fig. 9) with the CALIOP lidar data shows comparable results with respect to the PDF maxima location for ice ($T-T_{ICE}$ ~ -1 to 0 K) and a systematic shift to warmer temperatures for the other PSC classes. However, the shift for example for the CALIOP STS events is only in the range of 1-2 K, but the MIPAS analysis shows a shift of ~4 K. In addition, the CALIOP T-TICE histograms show significantly smaller PDF distribution widths than the MIPAS analysis.*

A quantified comparison between the MIPAS and CALIOP $T-T_{ICE}$ PDFs is partly affected by the different meteorological datasets used for both analyses (GEOS-5 for CALIOP and ERAi for MIPAS). Although the temperature biases in the winter polar region for most meteorological datasets have substantially improved over the last decade (Lambert and Santee, 2018), there are still significant differences in mean temperature biases between various assimilation systems. Regarding this comparison, this effect has the right tendency, with a warmer bias for ERAi than for GOES-5 (~0.5 K, taken radio occultation measurements for reference, Lambert and Santee, 2018). However, the different biases are not able to explain to the full extent the difference of 2-3 K of the $T-T_{ice}$ distributions and may also effect the maximum of the ice distribution in a similar manner.
A specific restriction of the MIPAS analysis is the caveat that NAT clouds with large particles (r > 2-3 microns) are difficult to distinguish from STS. The spectral characteristic spectral signature of NAT particles at 820 cm$^{-1}$ is for larger radii. As a consequence, these clouds may be misclassified as STS. This may cause a broader PDF and a shift to warmer temperatures (considering a difference in existence temperatures of $T_{NAT}-T_{STS}$ ~ 2-3 K).
Furthermore, due to a lack of coincident gas phase $HNO_3$ and $H_2O$ measurements for MIPAS in PSCs, it is unfortunately not possible to quantify their effects in detail, like in Pitts et al. (2013). Finally, the missing coincident $H_2O$ measurements may also create uncertainties in the $T_{ICE}$ estimates.

3) I am familiar with trying to identify PSCs from a precursor dataset to that of MIPAS, i.e., from the Nimbus 7 LIMS experiment. For the LIMS profiles I was able to determine the occurrence of PSC-like emission signatures as a function of threshold and/or existence temperature, but not its composition or type (see Remsberg and Harvey, Atmos. Meas. Tech., 2016). Your Figure 4 indicates that the predominant PSC type for NH winter is STS with its temperature threshold of ~192 K. I found a similar threshold temperature for PSC occurrences from LIMS, but could not verify that it was most likely STS based on Pitts et al. (2013, Fig. 8). Information in the literature at that time about likely composition was not as precise as you imply, in my opinion, and may still not be. Some guidance on this point would be welcome.

It is the strength of the MIPAS data, due to its spectral resolution and coverage, to identify and separate characteristic spectral signatures in the IR spectra. Consequently, the classification is temperature-independent and very robust, especially for small NAT and ice particles. But there is the caveat to miss-classify large NAT particles as STS, which is highlighted frequently in the manuscript (line 267, 635, or 676). In a coincidence comparison with CALIOP (Spang et al., 2016) we found for the MIPAS STS class large contribution of the CALIOP mix-types classes (STS+NAT). Only around 20% of the CALIOP coincidences were classified as 'pure' STS events. PSCs detected by MIPAS and classified with STS might be mixtures of STS and large NAT particles.

**References:**

Pitts, M. C., Poole, L. R., Lambert, A., and Thomason, L. W.: An assessment of CALIOP polar stratospheric cloud composition classification, Atmos. Chem. Phys., 13, 2975-2988, https://doi.org/10.5194/acp-13-2975-2013, 2013.

Spang, R., Hoffmann, L., Höpfner, M., Griessbach, S., Müller, R., Pitts, M. C., Orr, A. M. W., and Riese, M.: A multi-wavelength classification method for polar stratospheric cloud types using infrared limb spectra, Atmos. Meas. Tech., 9, 3619-3639, doi:10.5194/amt-9-3619-2016, 2016.

---

## Author Comment (AC2) · 9 Mar 2018

Reply to reviewer#2

We thank the reviewer for the helpful comments and suggestions on the manuscript. Please find below the point-by-point response and the changes in the manuscript. Replies are presented using times roman fonts and new or reworded text passages from the revised version are highlighted in *italic*.

Minor Comments:

Line 59: I guess not only in the Arctic this is important. . .

Like also reviewer #3 recommended, we are now highlighting Antarctica as well.

Paragraph starting line 83: Your statement '. . .are so far only very limited. . .' when discussing the current (prior to this MIPAS paper) state of PSC type observations. I would argue strongly that the previously-derived CALIOP satellite PSC Climatologies render this statement incorrect. See papers by Pitts et al. (ACP, 2007, 2009 etc). I recommend you be up-front about the previous CALIOP satellite climatologies and note its achievements and strengths, but then state clearly where your new MIPAS dataset fills in gaps left by CALIOP, and the MIPAS dataset strengths. This could be done in a new paragraph following line 110.

We followed the reviewer suggestion and rephrased this section including related Pitts et al. references as well and more context on MIPAS:

*CALIOP analyses of multiple winters of PSC particle type distributions in a climatological sense are presented by Pitts et al. (2009, 2011, and 2013). These studies stressed the scientific potential comprehensive multiannual PSC climatologies would have. PSC analyses of MIPAS were so far restricted on individual winters (Spang et al. 2005a, 2005b) or specific case studies (Höpfner et al., 2006b, Eckermann et al., 2009). A better latitudinal coverage up to the pole than CALIOP as well as homogenous day and night time coverage by MIPAS provide substantial additional information on climatological PSC distributions. Furthermore, MIPAS data are available for the time period from July 2002 to May 2006, before CALIOP became operational.*

Line 342: Provide a reference that SH winter 2010 was a warm winter. (de Laat & van Weele, Scientific Reports, 2011 , doi:10.1038/srep00038).

We followed the reviewer suggestion.

Figure 5: MIPAS NAT in 2009 (middle top figure) shows NAT between mid May and mid June below about 17km altitude. This is not seen in the CALIOP NAT area (middle bottom figure). This is separate from the tropopause-level NAT &ice issues which you discuss in the text. Can you explain why MIPAS sees NAT here but CALIOP doesn't?

We explained now more detailed the potential processes behind this artificial high cloud detection by both CALIOP and MIPAS in the following paragraph:
*"... for each PSC type in Figure 5. Note that the CALIOP and MIPAS detection algorithms are also sensitive to cirrus clouds in the tropopause region. In May, CALIOP and especially MIPAS show clouds classified as NAT at 12 km, which might be an artefact of the algorithms. It should be mentioned that CALIOP is detecting by far more ice than NAT clouds in this altitude region, and that the total coverage for NAT plus ice of both instruments is in good agreement. This is indicating that cirrus clouds at the tropopause are the most likely explanation for these detections. For so far unknown reasons these early winter cloud events at the tropopause are in most cases miss-classified as NAT for MIPAS. In addition, the MIPAS measurements show cloud detections at higher altitudes than CALIOP (up to 16 km) for this early winter period. The large FOV of MIPAS likely causes these unexpected and potentially overestimated cloud tops at and above the polar tropopause. Optical thick cirrus clouds in the lowest part of the FOV create overestimation in cloud top height of up to 1.5 to 2 km for IR limb sounders (Spang et al., 2012, Spang et al. 2015). The vertical grid box size of 2 km for MIPAS compared to 180 m for CALIOP causes also a slight overestimation in the cloud top occurrence statistics for MIPAS."*

Line 529-530: You should refer to de Laat & van Weele (Scientific Reports, 2011), doi:10.1038/srep00038 who show anomalously warm temperatures in MLS data and discuss the role of minor SSWs in 2010. Remove and replace line 530 and the personal communication with reference to this paper instead.

We followed the suggestion of the reviewer.

Technical Corrections:

All technical corrections have been considered in revised version of the manuscript.

---

## Author Comment (AC3) · 9 Mar 2018

Reply to Reviewer#3:

We thank the reviewer for the helpful comments and suggestions on the manuscript. Please find below the point-by-point response and the changes in the manuscript. Replies are presented using times roman fonts. New or reworded text passages in the revised version are highlighted in *italic*.

Comments:

(43) "PSC... and are formed of particles that are classified into three types..." as the particles are of three types, but coexist, giving rise to PSC classes more than three.

We changed the text to: *"PSCs are located in the cold polar vortices in both winter hemisphere, and are formed of three types of particle, which can also coexist: ..."*

(56) maybe here is "typology" instead of (or in addition to) "concentration"
We changed accordingly: *"…, where the rates depend on surface area, particle, and typology of the particle (...)."*

(60) Here, for completeness, the authors may also wish to quote the extensive dehydration that PSCs can induce in Antarctica.

We added a sentence on this topic.

(80) ". . .faster. . ." unclear. Faster than what?

We modified the corresponding sentence:
*Further, although reaction rates have been adjusted in recent model studies (e.g. Wegner et al., 2012), there is still a substantial uncertainty on the rates of heterogeneous reactions on NAT (Carslaw et al., 1997, Wegner et al., 2012) which makes determining the type of PSC present in the atmosphere important.*

(169) ". . . with altitude, and with altitude. . ."

Repetition is now deleted!

(191) "600000 modelled spectra with varying PSC types:" here it is unclear to me whether this forward model account for external mixtures of the three PSC particles. It seems otherwise, also in view of the following discussion, but maybe this this should be made clearer here.

Mixed type spectra have been not modelled in the database. Mixed type clouds will create spectra with mixed spectral characteristics and will make it more difficult to establish separation lines / thresholds with the Bayesian classifier. We added a comment on this restriction to `pure' PSC types in the model calculations.

(372) Typo. Corrected.
(439) Typo. Rephrased.

(440-450) The comparison shows good agreement, but I had some difficulty following the reasoning that tries to explain the only evident discrepancies between the two datasets: NAT and ICE in the low altitudes at the beginning of the season. While the absence of ICE at low levels in MIPAS may be due to the vertical truncation of its data, and while the detection of NAT around 12 km is commented for both datasets, the increased presence of NAT in the MIPAS measures between 12 and 15 km seems to me not well addressed. The authors could similarly comment on that small discrepancy.

We agree with the concerns of the reviewers (#2 and #3) and revised the whole paragraph. The NAT and ice partitioning was not represented correctly in the former description.

(542) Unfortunately in fig. 8 the years 2007 and 2008 are displayed with shades of green too close for me to be able to distinguish them.

Colour code has been changed in the new version of the manuscript.

(543 and following) Here, and elsewhere, the authors focus their attention on a particular Arctic winter, rather than reporting the climatology of mean values. I do not agree with this choice, given that the particular winter is, as highlighted in the text, exceptional, and therefore not very representative. It is also true that, given the

high arctic variability, even the average values are not very significant, but this can be highlighted, thus commenting on the low representativeness of the average conditions, while reporting the climatology over the whole dataset and transferring the discussion on the specific winter 2011 in a supplement.

We considered this before we submitted the paper. As the reviewer points out, there are arguments for and against a multi-annual mean of NH PSC seasons. We already presented an extended list of items characterising the mean NH conditions in the submitted manuscript version, but without showing a figure. Consequently, we followed the reviewer suggestion and replaced the plots for the winter 2010/11 in Figure 12 with the mean 2002-2012 statistic and highlighted the limited representativeness in the corresponding section in the text. The very specific winter 2010/11 is still described in section 4.1.2 (Fig. 8), whereby we omit repetitive parts of the description for this specific winter in 4.4.2. We also think, that a presentation of only one the specific winter in a supplement is not adding substantial information to the paper, if the main finding are already summarised in other sections of the article.

(612) Here the author may also quote that, to a lesser extent, also the denitrification and dehydration play a role in the downward propagation of PSC occurrence.

We added a corresponding sentence:
*Dehydration and denitrification processes and the corresponding redistribution of $H_2O$ and $HNO_3$ over the course of the winter have also influence on the downward propagation in PSC occurrence.*

(631) See comment for (543 and following).
See reply above.

(685-689) This interesting difference between Arctic and Antarctic seem to be explained by an artefact, but I did not get the explanation in full. Is this difference arising because high, thin arctic clouds are warmer that the corresponding Antarctic ones, at the same altitude and with the same optical thickness? If so, the authors may consider to rephrase the paragraph to make such statement clearer.

We rephrased the paragraph and left out misleading statements:
*This difference may be caused by a signal to noise ratio issue for altitudes of ~30 km and above (Spang et al., 2004, 2012). At these altitudes the cold stratospheric temperatures yield only very weak radiance signals in the atmospheric window region (close to the detector noise level) used in the cloud index approach. Consequently, cloud index profiles start to get noisy above ~30 km and cloud detection becomes more difficult. This effect is stronger in the SH than in the NH, with a larger and colder polar vortex in the SH. This may cause an underestimation in PSC occurrence at ~30 km in the SH.*

(690-700) Interesting feature. Have the authors tried to apply their classification algorithm in non-polar, volcanically contaminated stratospheric regions and see whether there too, the algorithm recognizes presence of NAT?

So far we have not applied the algorithm for mid and tropical latitudes. The classification algorithm is guided by the modelled IR spectra where we applied temperature and trace gas profiles representative for polar winter conditions. Applications at low latitudes would need to take realistic background profiles into account and some adaptation of the classification approach would be necessary.

(701-762) The author may consider to shift the whole paragraph in a supplement, and to quote only the main result in the manuscript.

We prefer to leave this section in the main part of the manuscript. This part is highlighting potential application of the new PSC database and gives a hint to the limits of proxies used to estimate the ozone loss potential over an entire winter like $A_{PSC}$ or $V_{PSC}$. In combination with Section 4.2.4, where additional smoothing and vertical averaging show a rather good correspondence of the simple temperature based proxies with the MIPAS observations, Figure 13 is stressing the limitations such simple proxies have compared to real observations of PSC (see also change notes (756) and (806) below).

(711) ". . . overall MIPAS PSC. . ." all? Antarctic?
Changed accordingly

(795) ". . . would certainly. . ." I would use ". . . could. . ." as you don't know unless you try. . .
Changed accordingly

(803) this statement seem in conflict with an earlier one at (753). Maybe one of the two should be rephrased.

We changed the statements (753) and (803) highlighting that the smoothing and vertical integration of the quantity Amax results in a better agreement:

(756) *(c) the simple temperature-based method is not accurate enough to describe the occurrence of PSCs with respect to vertical distribution and temporal evolution over the winter.*
…
(806) *The results of the comparison between the temporally smoothed and vertically integrated maximum area of PSC coverage of MIPAS with the simple temperature based PSC proxies show that the overall winter evolution can be modelled reasonably well. Although, more detailed and less smoothed analyses for individual winter show significant differences (see Figure 13). However, a similar approach applied to the output parameters of global models could be a valuable tool to quantify the quality of PSC related processes in CCMs and GCMs.*

---

## Author Response (AR2)

Dear Editor,

Thank you for your pleasant handling and carefully reading of our manuscript. We have considered all technical corrections listed in your report and hope the manuscript is now ready for publication.

Best regards

Reinhold Spang